# Metabolic phenotyping reveals an emerging role of ammonia abnormality in Alzheimer's disease

Tianlu Chen[1,6], Fengfeng Pan[2,6], Qi Huang[3,6], Guoxiang Xie[4], Xiaowen Chao[1], Lirong Wu[1], Jie Wang[3], Liang Cui[2], Tao Sun[1], Mengci Li[1], Ying Wang[2], Yihui Guan[3], Xiaojiao Zheng[1], Zhenxing Ren[1], Yuhuai Guo[1], Lu Wang[5], Kejun Zhou[4], Aihua Zhao[1], Qihao Guo[2,7] ✉, Fang Xie[3,7] ✉ & Wei Jia[1,5,7] ✉

The metabolic implications in Alzheimer's disease (AD) remain poorly understood. Here, we conducted a metabolomics study on a moderately aging Chinese Han cohort ($n = 1397$; mean age 66 years). Conjugated bile acids, branch-chain amino acids (BCAAs), and glutamate-related features exhibited strong correlations with cognitive impairment, clinical stage, and brain amyloid-β deposition ($n = 421$). These features demonstrated synergistic performances across clinical stages and subpopulations and enhanced the differentiation of AD stages beyond demographics and Apolipoprotein E ε4 allele (APOE-ε4). We validated their performances in eight data sets (total $n = 7685$) obtained from Alzheimer's Disease Neuroimaging Initiative (ADNI) and Religious Orders Study and Memory and Aging Project (ROSMAP). Importantly, identified features are linked to blood ammonia homeostasis. We further confirmed the elevated ammonia level through AD development ($n = 1060$). Our findings highlight AD as a metabolic disease and emphasize the metabolite-mediated ammonia disturbance in AD and its potential as a signature and therapeutic target for AD.

Alzheimer's disease (AD) is a progressive and irreversible neurodegenerative disease with a global increase in prevalence[1]. The diagnosis of AD and its stages currently rely on clinical symptoms, neuropsychological tests, and specific pathological features such as Amyloid-beta (Aβ) and tau pathology[2]. However, AD exhibits significant heterogeneity in phenotype, pathology, and progression, making early detection and understanding its mechanisms crucial[3]. For example, some individuals may have moderate or significant Aβ deposition but show no signs of cognitive impairment, while others may experience subjective cognitive decline (SCD) up to two decades before objective evidence of mild cognitive impairment (MCI).

Blood-based biomarkers that are easily accessible and cost-effective have garnered attention for their potential to improve AD diagnosis, optimize disease-modifying strategies, and enhance our understanding of the disease[2]. Metabolomics, the study of small molecules in living organisms, has emerged as a promising approach to characterize pathological processes and identify blood biomarkers for AD[4–7]. In our previous studies[5,8–10], we utilized quantitative metabolomics to examine the associations between metabolites and AD markers as well as the link between AD progression and changes in microbial metabolites using serum samples from the Alzheimer's Disease Neuroimaging Initiative (ADNI) and Religious Orders Study and

[1]Center for Translational Medicine and Shanghai Key Laboratory of Diabetes Mellitus, Shanghai Sixth People's Hospital Affiliated to Shanghai Jiao Tong University School of Medicine, Shanghai 200233, China. [2]Department of Gerontology, Shanghai Sixth People's Hospital Affiliated to Shanghai Jiao Tong University School of Medicine, Shanghai 200233, China. [3]Department of Nuclear Medicine & PET Center, Huashan Hospital, Fudan University, Shanghai 200040, China. [4]Human Metabolomics Institute, Inc., Shenzhen 518109, China. [5]Department of Pharmacology and Pharmacy, University of Hong Kong, Hong Kong 999077, China. [6]These authors contributed equally: Tianlu Chen, Fengfeng Pan, Qi Huang. [7]These authors jointly supervised this work: Qihao Guo, Fang Xie, Wei Jia. ✉e-mail: qhguo@sjtu.edu.cn; fangxie@fudan.edu.cn; weijia2@hku.hk

Rush Memory and Aging (ROS-MAP) cohorts. Our findings suggested the bile acid-ammonia axis and the gut microbiome-bile acid-brain cholesterol axis as potential targets for the prevention and treatment of AD[11]. In addition, we and others have observed that metabolic profiles in cognitively normal (CN), MCI and AD individuals are affected by age, sex, and Apolipoprotein E-ε4 allele genotype (APOE-ε4)[12–15]. Considering these covariates is important when studying metabolic changes related to AD and can help explain variations in AD susceptibility and severity across populations[16,17].

In this study, we investigated the plasma metabolic profiles of a cohort comprising moderately aging Chinese individuals (n = 1397, mean age = 66 years), encompassing both preclinical and symptomatic AD stages (Fig. 1). We comprehensively identified and evaluated metabolic features associated with AD stages, cognitive impairment, and brain Aβ deposition within the entire cohort, as well as in stratified populations and independent cohorts. Our objective is to enhance and validate prior findings in a new cohort, contributing to an improved understanding of AD as a metabolic disorder.

## Results

### Study cohort

Plasma samples of 1397 individuals, including 487 cognitively normal (CN), 239 with subjective cognitive decline (SCD), 284 with mild cognitive impairment (MCI), and 387 with Alzheimer's disease (AD), were obtained from the Chinese Preclinical Alzheimer's Disease Study (C-PAS)[18]. The mean age of the participants was 66.2 years (standard deviation = 8.6), with 41.3% of them being younger than 65 years. The majority of participants were women (65.6%), and 28.8% were identified as APOE-ε4 positive (Table 1). All participants underwent comprehensive assessments including general cognitive tests and a battery of standardized neuropsychological tests. A subset of 421 participants underwent brain positron emission tomography (PET) scans using the 18F-florbetapir amyloid tracer (also called 18F-AV-45) within one month after blood sampling. Detailed descriptions on neuropsychological tests and clinical diagnosis are provided in supplementary information.

### BAs, BCAAs, and excitatory neurotransmitters were closely associated with AD

A total of 189 metabolites belonging to 12 types were quantitatively measured (Fig. 2a and Table S1). Partial least squares discriminant analysis (PLS-DA) was employed to analyze the concentrations of all metabolites. The centroid scores plot illustrated distinct alterations in metabolic profiles at various disease stages, with the severity of the disease corresponding to an increased distance from the cognitively normal (CN) profile (Fig. 2b). Among the 12 metabolite types, the integrated levels (Fig. 2c, PC1 scores of PCA derived from metabolites belong to each type) of 5 types, namely bile acids (BAs), branch chain amino acids (BCAAs), excitatory neurotransmitters, amino acids, and medium and long chain fatty acids, exhibited significant differences among the four clinical stages (ANOVA FDR < 0.05 using log-transformed data). In comparison to CN, the levels of three marker types, BAs, BCAAs, and excitatory neurotransmitters, exhibited significant alterations (post hoc Dunnett's test p < 0.05) across more disease stages than other metabolites (Fig. 2c). We also integrated levels of metabolite types by summing up concentrations of metabolites belonging to each type directly and confirmed that these three types were different in more stages than the other types compared to CN (Fig. S1). We further observed the levels of five sub-types belonging to these three types in 4 clinical stages and found that the level of conjugated BAs was the highest in AD and the lowest in CN (AD > MCI > SCD > CN) and the level of glutamate-related metabolites was the lowest in AD and the highest in CN (AD < MCI < SCD < CN; Fig. 2d).

### Conjugated BAs, BCAAs, and glutamate-related features displayed stage-specific and population-specific associations with AD

The three identified metabolite types (BAs, BCAAs, and excitatory neurotransmitters) involve 3 BCAAs, 8 excitatory neurotransmitters, and 18 bile acids. Based on them, 34 extended features were generated (Table S2), comprising concentration summations and ratios reflective

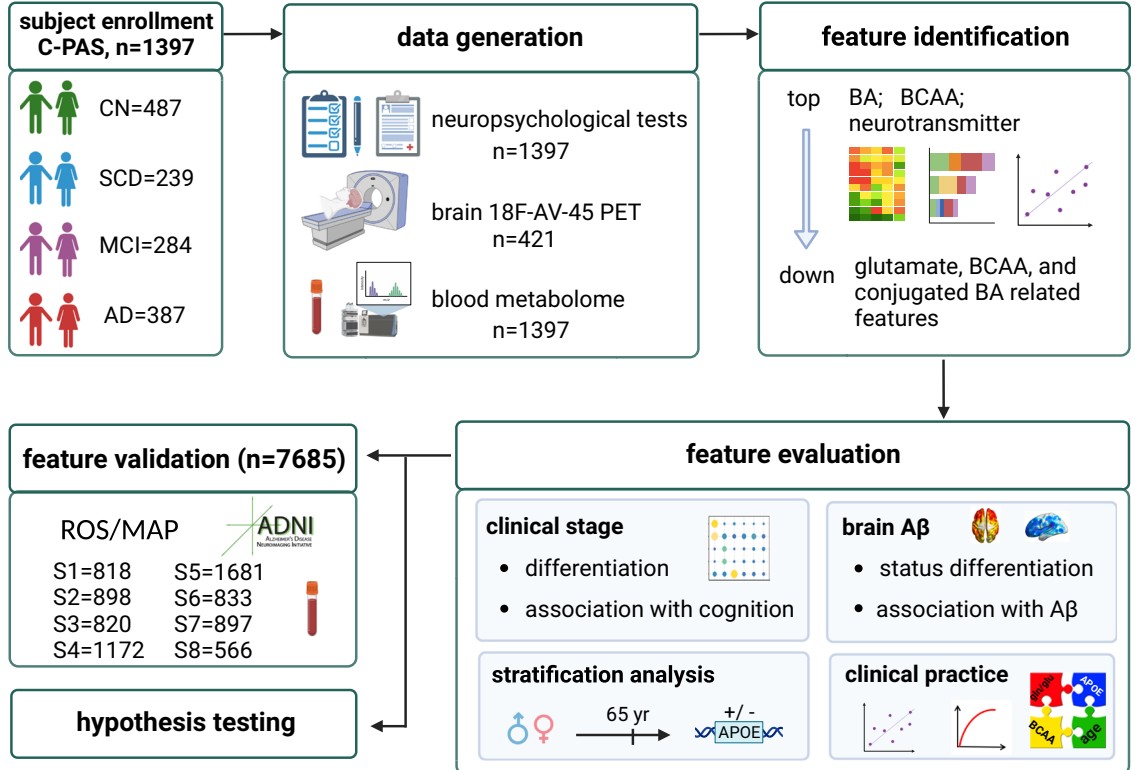

**Fig. 1 | Workflow of the study.** The study includes five steps: subject enrollment, data generation, feature identification, feature evaluation, and feature validation.

**Table 1 | . Characteristics of study population from C-PAS cohort**

| Characteristics | ALL (n = 1397) | CN(n = 487) | SCD(n = 239) | MCI(n = 284) | AD(n = 387) |
|---|---|---|---|---|---|
| Age (yr) | 66.2 + 8.6[a] | 63.6 + 8.2 | 64.4 + 7.4 | 66.5 + 8.1[b] | 70.4 + 8.4[b] |
| | [40,89] | [40,84] | [47,81] | [43,86] | [41,89] |
| | 66 (60, 72) | 64 (58, 69) | 64 (58, 70) | 66 (61, 73) | 71 (65, 77) |
| Sex (Men%) | 34.4% | 33.7% | 28.9% | 34.5% | 38.8% |
| BMI (kg/m$^2$) | 23.3 + 3.8 | 23.4 + 3.4 | 24.0 + 3.5 | 23.1 + 3.3 | 22.9 + 3.4 |
| | [13.7, 33. 8] | [15.4, 33.8] | [16.4, 31.6] | [15.5, 33.2] | [13.7, 31.1] |
| | 23.2 (21.0, 25.4) | 23.3 (21.2, 25.3) | 23.7 (21.5, 26.0) | 23.0 (20.9, 25.4) | 22.7 (20. 6, 25.2) |
| Education (yr) | 11.4 + 3.2[a] | 12.4 + 3.1 | 11.9 + 3.1 | 11.1 + 3.0 | 10.2 + 3.2[b] |
| | [6,22] | [6,22] | [6,20] | [6,22] | [6,19] |
| | 11 (9, 14) | 12 (10, 15) | 12 (9, 14) | 11 (9, 12) | 10 (7, 12) |
| *APOE* (ε4) carrier %[a] | 28.8%[a] | 19.1% | 17.6% | 29.6%[b] | 47.6%[b] |
| PET acceptance(%)[b] | 30.1%[a] | 34.8% | 39.3% | 31.7%[b] | 17.3%[b] |
| Brain Aβ + (%)[c] | 28.5% | 15.3% | 18.1% | 31.1% | 73.1%[b] |
| MMSE | 24.6 + 5.6[a] | 28.2 + 1.7 | 27.7 + 1.8[b] | 26.5 + 2.1[b] | 16.8 + 4.7[b] |
| | [10,30] | [20,30] | [21,30] | [15,30] | [10,27] |
| | 27 (22, 29) | 28.5 (27, 29) | 28 (26, 29) | 27 (25, 28) | 17.5 (12, 21) |
| ACEIII-CV | 68.8 + 18.2[a] | 82.0 + 7.9 | 77.8 + 8.0[b] | 70.3 + 9.0[b] | 45.7 + 14.7[b] |
| | [10,97] | [60,97] | [60,96] | [50,94] | [10,77] |
| | 73 (60, 82) | 83 (77, 88) | 78 (73, 83) | 71 (64, 76) | 48 (36, 58) |
| MoCA-BC | 23.3 + 4.8[a] | 26.1 + 2.5 | 24.7 + 3.0[b] | 21.9 + 3.4[b] | 15.2 + 3.3[b] |
| | [10,30] | [20,30] | [17,30] | [15,30] | [10,22] |
| | 24.50 (20, 27) | 27 (25, 28) | 25 (22, 27) | 22 (20, 24) | 15 (12, 18) |

Data are presented as mean+S.D., [minimum, maximum], and median (IQR), or percentage.
[a]indicates Chi-squared test, analysis of variance, or Kruskal–Wallis test FDR < 0.05 when comparing 4 groups (adjusted by Benjamini and Hochberg).
[b]indicates Chi-squared test, student's t-test or Mann-Whitney test FDR < 0.05 when compared to CN (adjusted by Benjamini and Hochberg). *C-PAS* Chinese Preclinical Alzheimer's Disease Study, *CN* Cognitively normal, *AD* Alzheimer's disease, *SCD* Subjective cognitive decline, *MMSE* Mini-Mental State Examination, *ACEIII-CV* Chinese version of Addenbrooke's cognitive examination-III, *MoCA-BC* Chinese version of Montreal Cognitive Assessment-Basic. a: the percentage of *APOE-ε4* carriers. b: the percentage of the participants that accepted brain PET test. c: the percentage of participants with positive Aβ (defined through visual assessment following the guidelines for interpreting amyloid PET) in those underwent the brain AV45-PET scans.

of enzymatic activities or gut microbiome function[19–22]. We constructed linear regression models (age, sex, BMI, *APOE-ε4*, and education year were adjusted) and identified 13 features (out of 63 comprising 29 metabolites and 34 extended features) significantly associated (FDR < 0.05) with clinical stages within the entire population (refer to the first column in Fig. 3a). Notably, six features related to BAs (five of which were linked to conjugated BAs), four to BCAAs, and three to excitatory neurotransmitters (all involving glutamate) exhibited associations. Positive associations (positive effect sizes) with disease severity were observed for BA-related features, whereas BCAA and glutamate-related features displayed negative associations (negative effect sizes) with disease severity. Subsequently, logistic regression models were constructed on these 13 features, facilitating the differentiation of every two stages (M1-M6) while adjusting for covariates. Interestingly, the number of features with different levels ($p < 0.05$) was smaller in SCD and MCI (M4), while CN and AD (M3) exhibited a larger disparity than other comparisons within the entire population (Fig. 3a). The stage-specific and complementary performances of these features were evident, as distinctions were observed in BA-related and glutamate-related features across all comparisons, except for M4 (SCD vs. MCI) and M1 (CN vs. CSD) respectively while BCAA-related features showed differences between M1 (CN vs. SCD) and M3 (CN vs. AD). Furthermore, the association patterns of these features with cognition were evaluated (Fig. 3b). Comparatively, glutamate-related features exhibited the highest number of associations, followed by BA- and BCAA-related features.

Stratification analysis, accounting for age, sex, and *APOE-ε4* characteristics, revealed diverse patterns across sub-populations (Fig. 3c–e). *APOE-ε4* status exerted a more substantial impact on these features compared to sex and age. A lower number of differential features was observed among four stages (Fig. 3f) and in pairwise comparison between clinical stages (Fig. 3g), as well as in features associated with cognition (Fig. S3) among *APOE-ε4* carriers compared to non-carriers. Men exhibited a higher number of BA-related features than women across four stages (Fig. 3f), between two stages (Fig. 3g), and in associations with cognition (Fig. S3). Conversely, women and younger participants showed a greater prevalence of BCAA-related features in these scenarios.

**Glutamate metabolism was associated with brain Aβ deposition**
Many lines of evidences support that Aβ plays a key role in AD pathology, and which is an established indicator of AD pathology. Associations between the 13 features and brain Aβ deposition were examined ($n = 421$). We observed consistent patterns in the levels of glutamate-related features across different populations. Specifically, glutamate-related features were consistently lower in individuals with positive Aβ (Aβ + ) compared to those with negative Aβ (Aβ-) in CN + SCD groups ($n = 264$), MCI + AD groups ($n = 157$), and the entire study population (Fig. 4a–c). However, it's important to note that these differences did not reach statistical significance in the CN + SCD group alone, with $p$-values exceeding 0.05 in both the Mann-Whitney test and logistic regression adjusting for age, sex, BMI, education year, and *APOE-ε4* status. Despite this, we identified a noteworthy finding in CN + SCD subjects, where GDCA/DCA was significantly higher (Mann-Whitney $p = 0.0082$) in Aβ+ individuals ($n = 43$) compared to Aβ- individuals ($n = 221$). This suggests an early alteration in individuals without clinical symptoms but exhibiting signs of AD pathology. The significance of GDCA/DCA was attenuated after adjusting for covariates (logistic regression adjusting for the aforementioned covariates; $p = 0.073$).

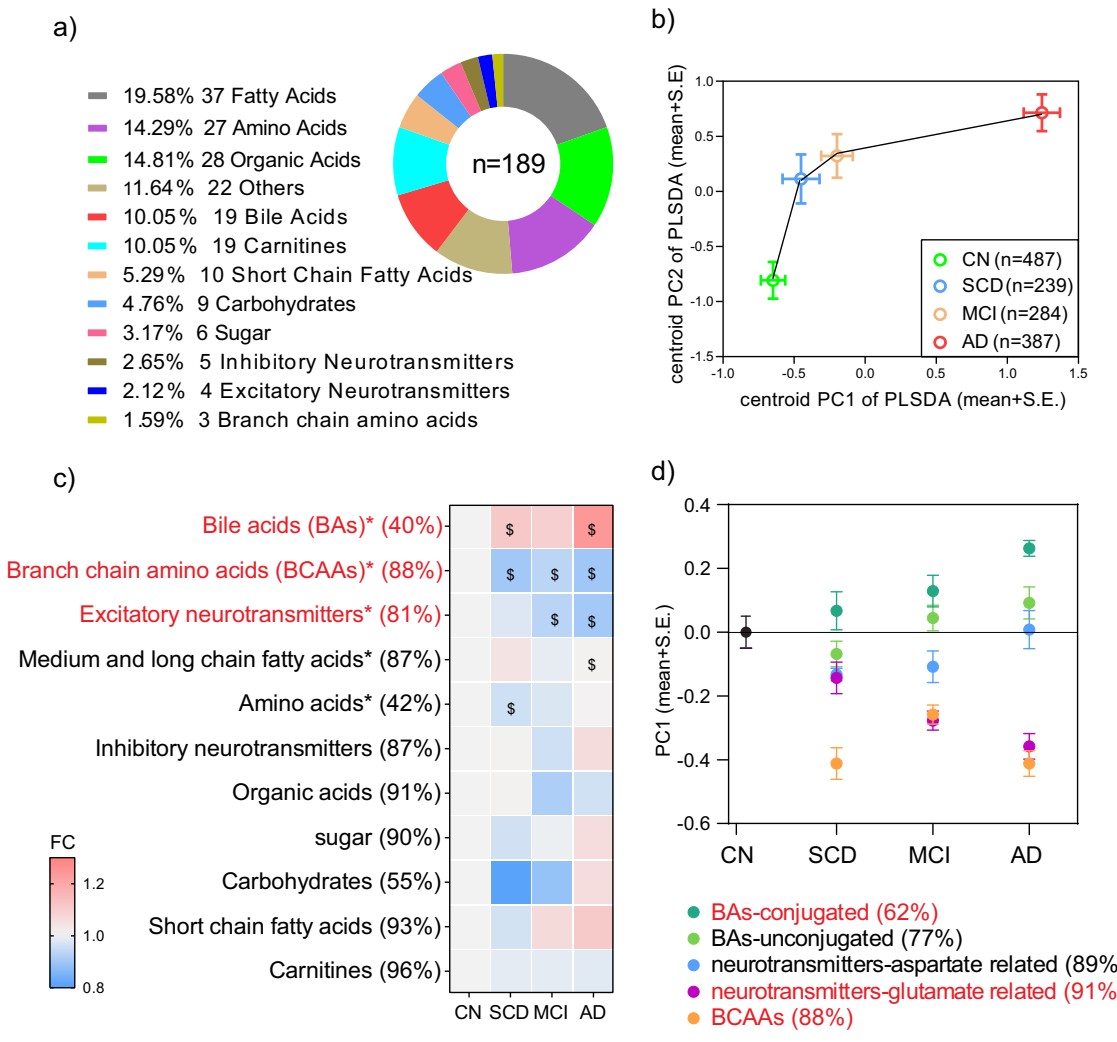

**Fig. 2 | BAs, BCAAs, and excitatory neurotransmitters exhibit strong associations with AD. a** Composition of metabolic profiles stratified by metabolite type. **b** The partial least squares discriminant analysis (PLS-DA) scores plot generated by the raw concentrations of all metabolites and samples. The circles and error bars represent the centroids (mean+S.E.) of principal component scores corresponding to each stage. **c** Fold changes of PC1s (the first component of PCA) derived from the metabolite types in subjects with SCD, MCI, and AD relative to CN. * indicates ANOVA FDR < 0.05 (two-sided) when comparing NC, SCD, MCI, and AD. $ indicates post hoc Dunnett's test $p < 0.05$ (two-sided) when compared to CN. Metabolite types are arranged in decreasing order of ANOVA FDR values. The number next to the name represents the percentage of variation that PC1 captured. **d** Levels (mean with S.E.) of five sub-types belonging to the top three types of (**c**) in four clinical stages. The levels are represented by the PC1 scores derived from PCA based on metabolites belonging to each sub-type and were scaled to the same starting point ($n = 487, 239, 284, 387$ for CN, SCD, MCI, and AD respectively). The number next to the name represents the percentage of variation that PC1 captured. Source data are provided as a Source Data file.

Moreover, employing voxel-wise analysis, we delved into the associations between the identified features and brain Aβ deposition. Our results confirmed a negative association between glutamate-related features, particularly glutamate/glutamine, and brain Aβ deposition, predominantly in the frontal, lateral parietal, and lateral temporal lobes (Fig. 4d–t). This underscores a close correlation between the glutamate/GABA-glutamine cycle and AD pathology. Interestingly, this correlation exhibited some dependency on factors such as age, sex, *APOE-ε4* status, and disease stage.

### Metabolic features enhanced clinical markers for the associations with clinical stages and brain Aβ deposition

The clinical contributions of the identified features were evaluated by comparing the performances of gradient boosting models based on basic markers (age, sex, BMI, education years, and *APOE-ε4* status) and models incorporating the 13 features. The inclusion of metabolic features consistently improved the auROCs (all higher than zero with an average of 0.05 and a maximum of 0.13) of models for stage

differentiation and prediction (Fig. 5a–g). The improvement in the correlation coefficient with global Aβ deposition ([18 F]florbetapir SUVr) increased from 0.21 to 0.47 (Fig. 5h).

### Replication of feature performance in matched samples and other data sets

Given the importance of age, we examined the performance of the 13 features in an age-matched sub-population (Table S3, $n = 991$, mean age = 69.7, ranging from 60 to 89), despite age adjustments being made in the above analyses. In line with the entire population, BA-related features showed positive associations, while BCAA and glutamate-related features exhibited negative associations with disease severity (Fig. S3a). Glutamate-related features displayed the highest number of associations with cognition, followed by BA- and BCAA-related features (Fig. S3b). Sex and *APOE-ε4* stratified analyses indicated that *APOE-ε4* status had a more substantial impact than sex (Fig. S3c, S3d).

The performances of the 13 features were further verified in eight data sets comprising a total of 7685 participants (Tables S4 and S5).

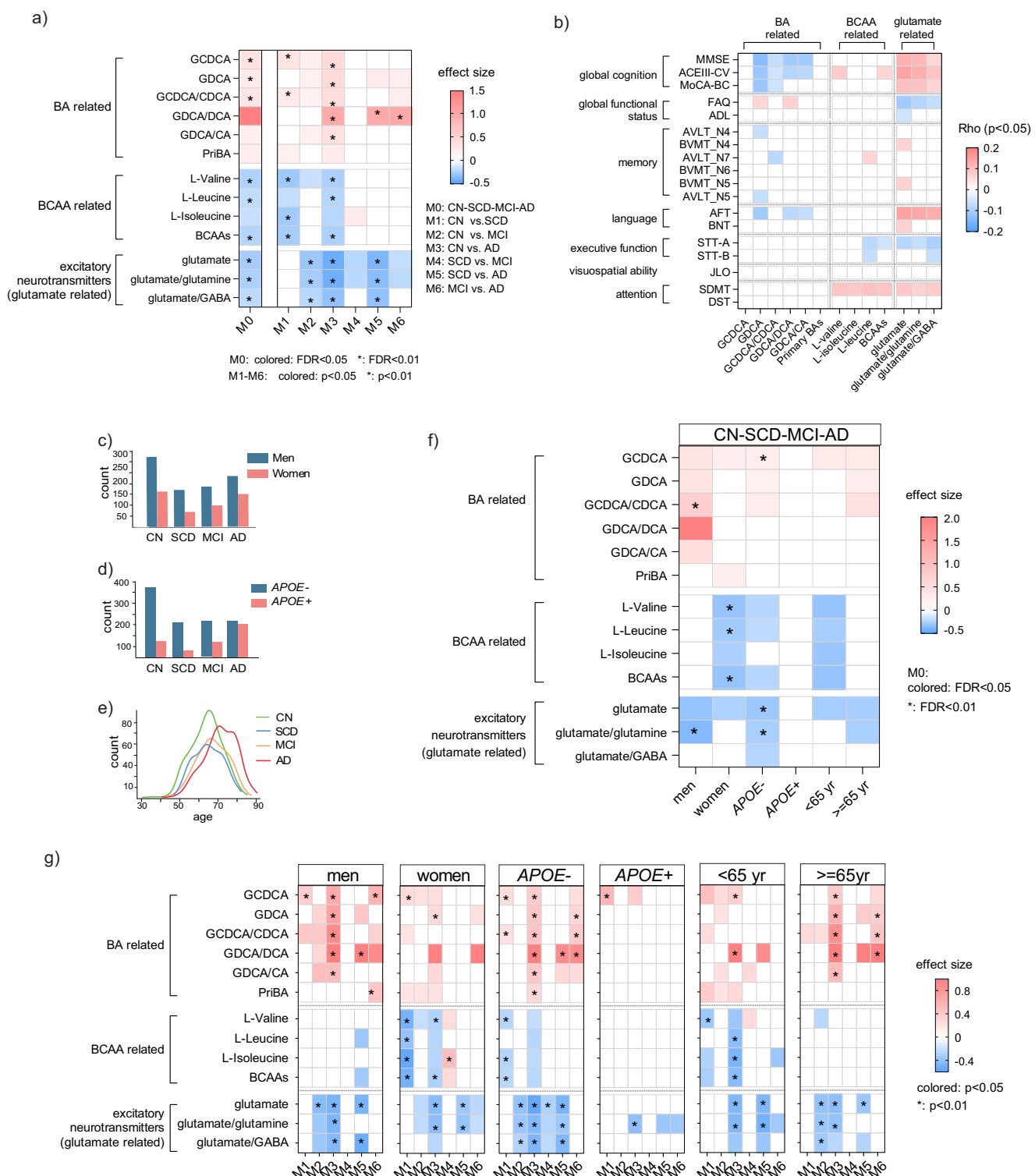

Given the overlap of samples in ADNI data sets (Figure S4), we employed a three-level meta-analysis to evaluate their differences between CN and AD and their associations with global cognition. Detailed results for each feature are provided in Figures S5 and S6. The majority (11 out of the 13) features showed significant alterations in AD compared to CN (Fig. 6a, $p < 0.05$, random effect model), and their alteration trends were consistent with the findings from the C-PAS cohort. The overall results on Spearman correlation coefficients between metabolic features and global cognition scores (Fig. 6b) indicated that 11 among the 13 features were associated with cognition

($p < 0.05$). Consistent with the results of C-PAS, BA-related features were positively and BCAA- and glutamate-related features were negatively associated with cognition decline.

## Metabolite-mediated ammonia abnormality in AD

The identified features were found to be associated with elevated blood ammonia levels, which is a neurotoxin and has been implicated previously in AD pathology (Fig. 7a). Analysis of available plasma samples from the C-PAS cohort, $n = 1060$, including 329 CN, 198 SCD, 230 MCI, and 303 AD, confirmed that ammonia levels were the highest

**Fig. 3 | Conjugated BAs, BCAAs, and glutamate-related features were associated with clinical stages and cognition in a stage-/population-specific way. a** Effect sizes of 13 metabolic features significantly different among four clinical stages (M0) and their performances in differentiating every two stages (M1-M6) based on linear regression (M0) and logistic regression models (M1-M6) respectively. Data from entire population was used. Colored cell indicates FDR < 0.05 (M0) or $p < 0.05$ (M1-M6) and * indicates FDR < 0.01 (M0) or $p < 0.01$ (M1-M6). Covariates including age, sex, BMI, education year, and *APOE-ε*4 were adjusted. The $p$-values were based on two-sided tests. **b** Associations of 13 metabolic features and cognition scores (entire population). Cell color indicates the correlation coefficient from Partial Spearman analysis (red: positive; blue: negative; blank: $p \geq 0.05$; two-sided). **c** The distribution of men and women in four clinical stages. **d** The distribution of *APOE-ε*4 carriers and non-carriers in four clinical stages. **e** Age distribution of participants in four clinical stages. **f** Effect sizes of 13 metabolic features when differentiating four clinical stages based on linear regression models (M0) in stratified populations. Colored cell indicates FDR < 0.05 and * indicates FDR < 0.01

(two-sided). **g** Effect sizes of 13 metabolic features when differentiating CN and SCD (M1), CN and MCI (M2), CN and AD (M3), SCD and MCI (M4), SCD and AD (M5), and MCI and AD (M6) respectively in stratified populations. Colored cell indicates $p < 0.05$ and * indicates $p < 0.01$ (two-sided). The correlation coefficients of 13 metabolic features and cognition scores based on Partial Spearman in stratified populations are shown in Figure S2. Source data are provided as a Source Data file. GCDCA Chenodeoxycholic acid glycine conjugate, GDCA Deoxycholic acid glycine conjugate, CDCA Chenodeoxycholic acid, DCA Deoxycholic acid, CA Cholic acid, PriBA concentration summation of primary BAs, MMSE Mini-Mental State Examination, ACEIII-CV Chinese version of Addenbrooke's cognitive examination-III, MoCA-BC Chinese version of Montreal Cognitive Assessment-Basic, FAQ Functional Assessment Questionnaire, ADL Activities of Daily Living, AVLT Auditory Verbal Learning Test, BVMT Brief Visuospatial Memory Test, AFT Animal Verbal Fluency Test, BNT Boston Naming Test, STT Shape Trail Test, JLO Judgement of Line Orientation, SDMT Symbol Digit Modalities Test, DST Digit Span Test.

in AD and the lowest in CN (AD > MCI > SCD > CN; Fig. 7b) and were negatively associated with cognition and BCAA- and glutamate-related features, while positively associated with BA-related features (Fig. 7c). These findings highlight the contributions of BA, BCAA, and glutamate metabolism to peripheral ammonia levels and their combined impact on AD development.

## Discussion

Previous studies have shown strong evidence for the associations between blood-based metabolic features and AD[1,23]. However, most of these studies focused on European ancestry populations, with only a few studies conducted in East Asians. In our previous research, we identified and proposed potential roles of altered metabolic features in AD development, such as regulating gut microbiota and brain cholesterol catabolism, using populations from the United States and China[5,8–10]. In this study, we examined the plasma metabolome of a new Han Chinese cohort, including relatively young participants and individuals at the preclinical stage of AD. We not only confirmed previously reported associations of conjugated BAs, BCAAs, and glutamate metabolism with AD progression but also extended our analysis to assess their performances in stratified populations and independent cohorts. Beyond their connections to clinical stages and cognitive function, we identified the association of glutamate-related features with AD pathology. Furthermore, we proposed the intermediary role of ammonia in the relationship between metabolic features and AD, providing additional support to the emerging hypothesis that ammonia disturbance contributes to AD progression. Our findings significantly contribute to expanding our comprehension of AD as a metabolic disease, offering observational evidence for early detection and advancing our understanding of AD pathology.

The metabolic profiles were different among clinical stages and were associated with disease severity. Features related to conjugated bile acid, BCAA, and glutamate metabolism were extensively examined in entire and stratified populations. We observed that their performances were not only stage-specific but also population-specific. This implies that employing distinct sets of metabolic features tailored to specific stages and populations enhances the precision in characterizing the metabolic patterns of AD. These observations may also help explain the inconsistencies in associations between metabolites and AD reported in other studies, and aid in interpreting the differences in pathophysiology and symptoms observed across different populations. On the other hand, caution is warranted in interpreting our findings. Further analysis, incorporating additional covariates and exploring their interactions, will contribute to a more nuanced understanding of these features and the overall metabolic landscape in AD. More in-depth discussions on the potential influences of age, sex, and *APOE-ε*4 status on the association between metabolic profiles and the progression of AD are provided in supplementary information.

The alterations in the identified features align with previous cross-sectional and longitudinal reports[6–9,24–27]. These features have been proposed to have direct or indirect roles in various processes linked to AD pathology, including ammonia abnormality, gut microbiome disturbance, energy metabolism disorder, mitochondrial dysfunction, oxidative stress, apoptosis, and neuronal autophagy. Our findings further support the emerging role of metabolite-mediated ammonia disturbance in AD, as the identified features are involved in the regulation of blood ammonia homeostasis through several mechanisms (Fig. 7a). The higher levels of conjugated BA-related features in AD can be attributed to enhanced BA reabsorption in the intestine and increased transport of conjugated BAs into the bloodstream. This leads to a decrease in conjugated BAs in the intestinal tract, resulting in increased ammonia levels. The observed lower levels of glutamate and glutamate/glutamine in AD and MCI compared to CN suggests abnormalities in the glutamate-glutamine cycle, which generates ammonia as a byproduct[28]. Ammonia has significant effects on glutamatergic and GABAergic neuronal systems, which are predominant in cortical structures[29]. BCAAs are connected to ammonia through their conversion to glutamate[30] and their involvement in protein synthesis and degradation[31]. Clinical trials have shown that oral administration of BCAAs can effectively reduce blood ammonia levels and benefit individuals with hepatic encephalopathy and impaired cognition[32]. While BCAAs are recognized risk factors for insulin resistance (IR) and type 2 diabetes (T2D)[33,34], which are independent risk factors for AD[35], their effects on IR/T2D and AD may differ. The core connection between IR/T2D and AD may not solely rely on the role of BCAAs. Together, existing evidence is rich but insufficient to fully elucidate the impact of identified metabolic features to AD progression. Ongoing investigations, incorporating longitudinal designs and accounting for factors like nonlinear changes, complications, medications, and diet, are underway.

Glutamate metabolism is associated with the pathological process of AD and blood levels of glutamate-related features may serve as a marker reflective of brain glutamate metabolism and Aβ deposition. Previous studies[30,36–38] have reported contradictory results (lower levels vs. higher levels) of glutamate in AD patients. Our findings of a negative relationship between glutamate levels in plasma and amyloid deposition in brain are inconsistent with the notion that elevated glutamate induces neurotoxicity, impacting neurons adversely[39]. Conversely, reduced glutamate levels could signify synaptic dysfunction and cognitive decline[40,41]. Diverse methodologies, patient heterogeneity, and disease progression stages may collectively contribute to the observed discrepancies. It is important to note that the current evidence is insufficient to fully determine whether this association is causal or a consequence of Aβ deposition. Additionally, as our study did not include other types of dementia, we cannot ascertain the specificity of these markers for

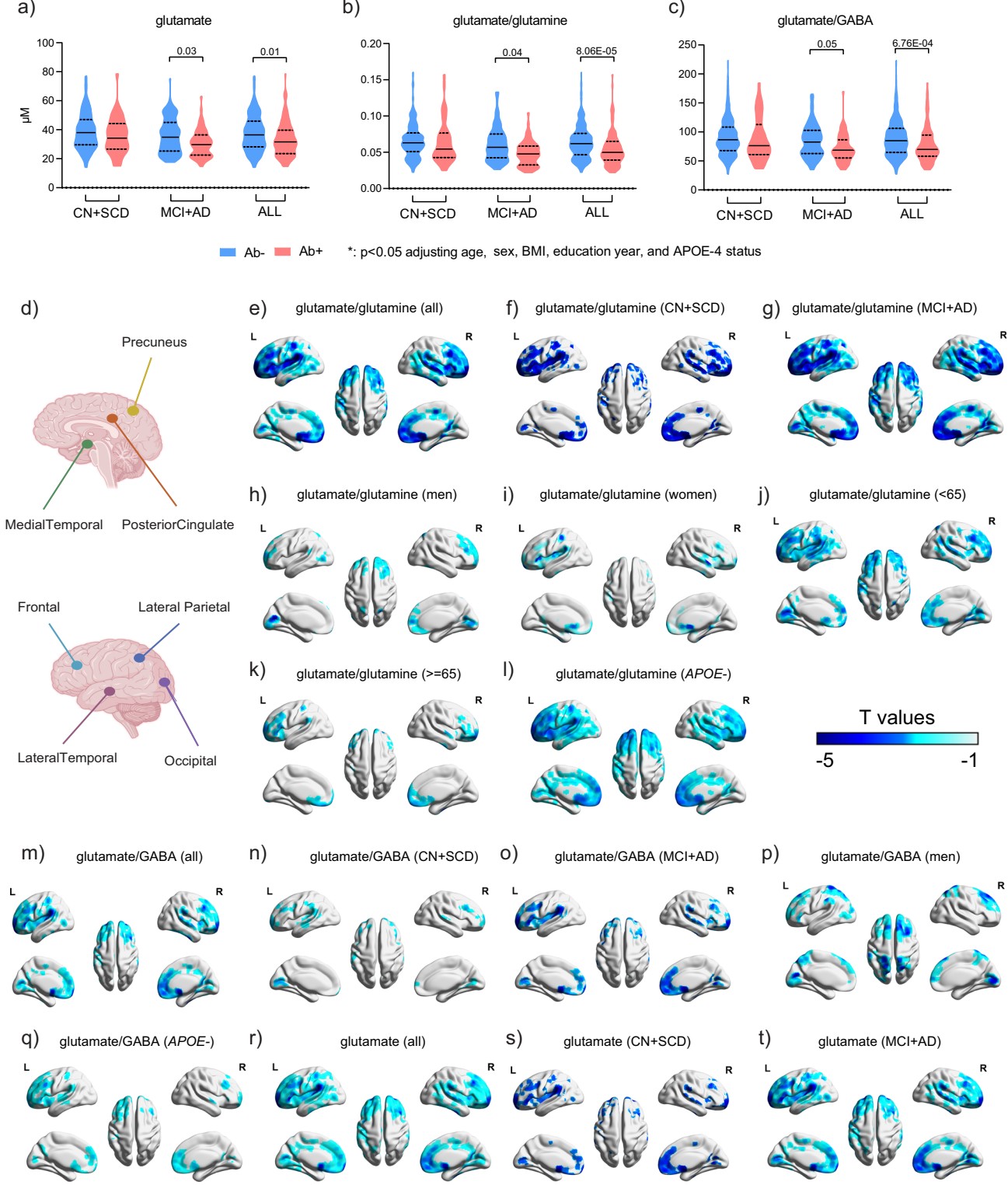

**Fig. 4 | The associations of glutamate metabolism with brain Aβ deposition (*n* = 421).** The levels of glutamate (**a**), glutamate/glutamine (**b**), and glutamate/GABA (**c**) in CN + SCD, MCI + AD, and all participants stratified by positive and negative Aβ deposition which was determined by the consensus of physicians' visual interpretation of PET image. The solid line in violine plot represents the median and the dashed line represents quartile. The p values were from Mann-Whitney test (two-sided). **d** Typical brain regions. Brain Aβ deposition were negatively associated with glutamate/glutamine (**e–l**), glutamate/GABA (**m–q**), and glutamate (**r–t**), based on the voxel-wise analysis, in entire and stage-/sex-/age-/*APOE*-ε4-stratified participants. The significance level of the linear regression (**e–t**) was set at *p* < 0.05 (two-sided) with peak-level false discovery rate (FDR) correction and the cluster-defining voxel threshold at the default of 0.001. The color bar stands for the T values of the voxel-wise analysis. Covariates including age, sex, BMI, education year, and *APOE*-ε4 were adjusted when applicable. Source data are provided as a Source Data file.

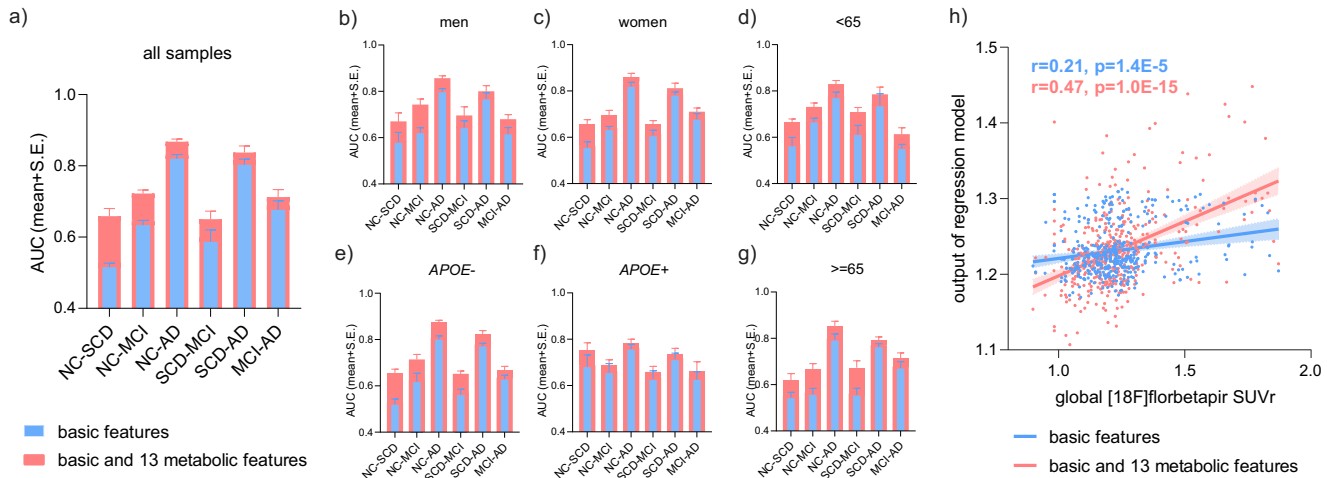

**Fig. 5 | Metabolic features improved the performances of clinical markers in entire and stratified participants.** The auROC values (mean with S.E. from 7-fold cross validation) of gradient boosting models using basic (age, sex, BMI, *APOE-ε4*, and education year; blue) and combined (basic and 13 metabolic features; red) features for the differentiation of every 2 stages, in all (**a**) and stratified (**b–g**) participants. The sample numbers for NC, SCD, MCI, and AD are 487, 239, 284, and 387 respectively, for all (**a**) and stratified analysis (**b**:165, 69, 99, 154; **c**: 322, 170, 185,

233; **d**: 255, 122, 110, 90; **e**: 393, 197, 200, 203; **f**: 94, 42, 84, 184; **g**: 232, 117, 174, 297). **h** Scatter plot of whole brain Aβ deposition level and output of gradient boosting regression model with (red, Spearman correlation coefficient r = 0.47, p = 1.0E-15, two-sided) and without (blue, r = 0.21, p = 1.4E-5, two-sided) metabolic features (n = 421). The lines are linear fitting lines with 95% CI. Source data are provided as a Source Data file.

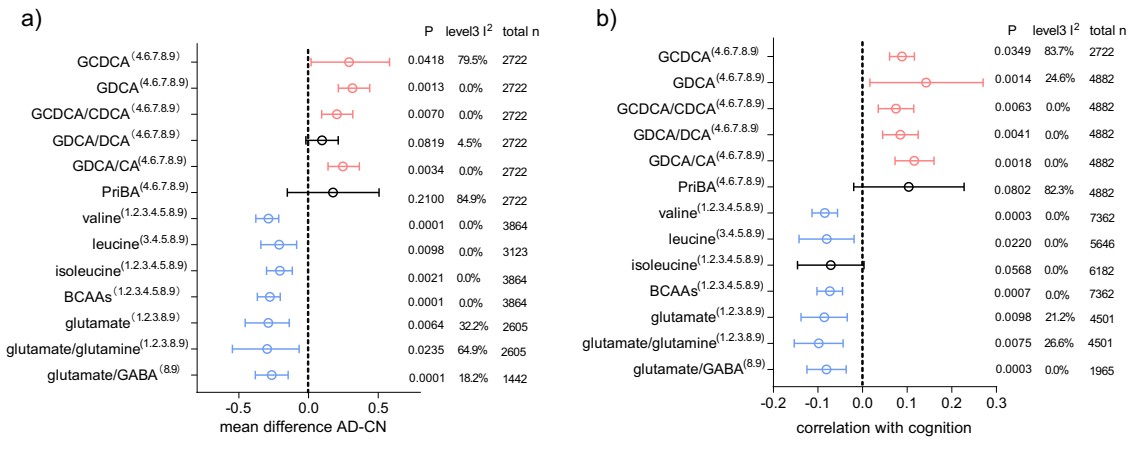

**Fig. 6 | Replication of feature performance in other data sets.** Three-level meta-analysis forest plots of identified features on their alteration trends with AD progression (**a**; standardized mean difference with 95%CI of AD and CN stages; > 0, higher in AD; < 0, higher in CN) and on their associations with global cognition (**b**; Partial Spearman r value with 95%CI based on all available data. Age and sex were adjusted.). The p-values were based on two-sided tests. C-PAS (study No.9) and another 8 data sets derived from ADNI and ROSMAP were involved. 1: ADNI-

Duke2016; 2: ADNI-Duke2017; 3: ADNI-California2017; 4: ADNI-Hawaii2021; 5: ADNI-Nightingale2021; 6: ADNI-DukeBAs2016; 7: ADNI-M2OVEAD2016; 8: Rosmap-Hawaii2017; 9: C-PAS-Shanghai2023. Variables were scaled to 0-1 within each study respectively and p-values were from random effect models. Cognition scores for ADNI, C-PAS, and ROSMAP data sets were ADAS-13, -1*MMSE, and a composite measure of global cognition created by averaging the z-scores of all tests respectively. Source data are provided as a Source Data file.

AD. Further investigations are warranted to elucidate the interplay of glutamate-related features in both the brain and blood, unravel their causal associations, and determine their specificity in relation to brain Aβ deposition.

Accurate and timely diagnosis of AD, including preclinical and prodromal stages, remains challenging for clinicians[1]. Currently, CSF and PET tests, which are costly and invasive, are not widely accessible. Therefore, there is increasing reliance on blood-based markers in the diagnostic workup of AD. Although plasma A/T/N markers have garnered attention, their levels in plasma are much lower than in the brain and CSF, posing challenges for current testing technology and hindering their clinical implementation. In comparison, the detection of the identified features in plasma is reproducible and cost-effective. Our data

demonstrate their consistent contributions to the differentiation of clinical stages, making them potential alternative markers in clinical practice, particularly in scenarios requiring frequent and long-term monitoring. We here propose a small panel of 4 features as promising biomarker targets considering their overall performances and underlying biological significance. They are GDCA/DCA, GDCA/CA, valine, and glutamate/glutamine.

The strengths of our study include the distinctive population, the top-down and stratification analysis approach, the imaging-based association analysis, and the validation in multiple cohorts. However, there are several limitations to consider. First, the findings are observational and await replication in longitudinal studies with larger sample sizes, appropriate stratification, and consideration of additional

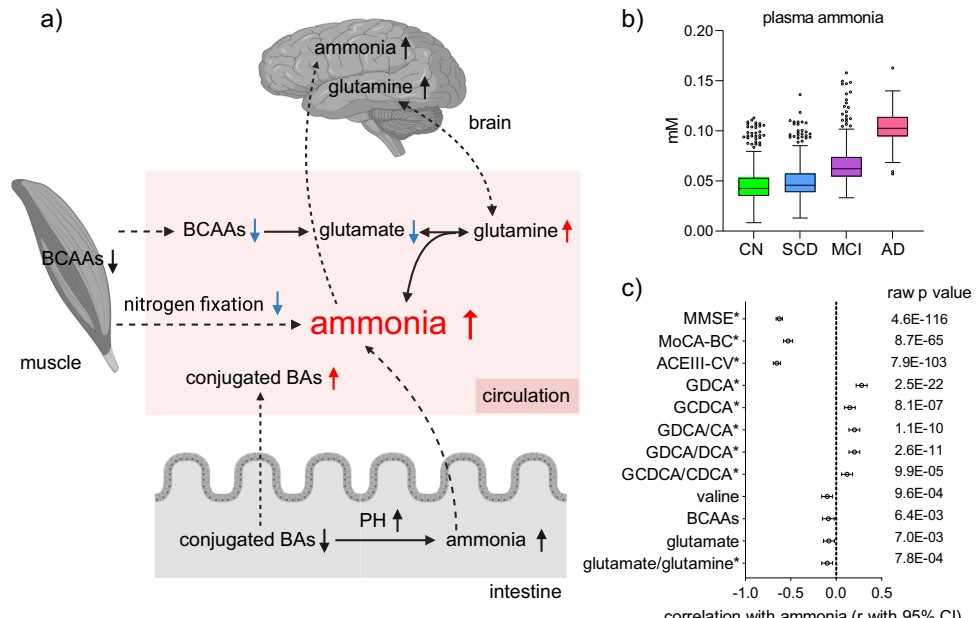

**Fig. 7 | Metabolite-mediated ammonia abnormality in AD. a** Biological associations of elevated blood ammonia and metabolic features related to conjugated BA, glutamate, and BCAA. **b** Levels of blood ammonia in four clinical stages. The sample number for CN, SCD, MCI, and AD are 329, 198, 230, and 303 respectively. The centre line denotes the median value (50th percentile), while the box contains the 25th to 75th percentiles of dataset. The whiskers mark the 5th and 95th percentiles, and values beyond these upper and lower bounds are considered outliers. **c** Associations (correlation coefficient r with 95% CI) between blood ammonia and cognition and metabolic features ($n = 1060$). Cognition scores and metabolic features with Spearman correlation $p < 0.05$ (two-sided) are listed. * represents $p < 0.05$ (two-sided) after adjusting age, sex, BMI, education year, and *APOE*-ε4. Source data are provided as a Source Data file.

risk factors and confounders. Further research is required to elucidate the collective impacts of identified features and ammonia on AD. Second, our top-down analysis may have overlooked sporadic metabolites whose significance may have been overshadowed by others within the same group. Employing two- or three-fold stratification analysis could enhance the granularity of the findings. Third, the subjects are relatively younger than that of some other AD cohorts and more patients with familial AD may be included. All the samples were collected in a non-fasting state. However, it is important to note that the performances of the identified features are consistent in some other cohorts with older subjects and fasting samples, which enhances the reliability of our findings.

In summary, leveraging the new Han cohort alongside several publicly available datasets, we validated the associations of conjugated BAs, BCAAs, and glutamate metabolism with AD. Our study unveiled the diverse performances of these metabolites in sub-populations, offering powerful evidence for the mechanistic link between metabolite-mediated ammonia abnormality and AD development. Future investigations are imperative to validate the clinical potentials of the identified features and unravel their interconnected roles in maintaining ammonia homeostasis, influencing Aβ deposition, and contributing to AD development. Further exploration in these directions holds promise for advancing our understanding and potential interventions in AD.

## Methods
### Study cohort and samples
The ethics committee of Shanghai Sixth People's Hospital Affiliated to Shanghai Jiao Tong University School of Medicine reviewed and approved the C-PAS Study (2019-032), following the principles of the Declaration of Helsinki. Written informed consent was obtained from participants or their caregivers. All relevant ethical regulations were followed during the study. ADNI is a multi-center study focused on biomarker development for early AD detection and tracking. Informed consent was obtained from participants, and the study was approved

by each participating site's institutional review board. The ROSMAP study was approved by the review board of Rush university, and participants provided informed consent.

**C-PAS**. Plasma samples and related information were obtained from 1397 individuals enrolled in the C-PAS from April 2019 to June 2021. C-PAS is a nationwide longitudinal study aimed at identifying biomarkers for early detection and progression tracking of Alzheimer's disease (AD). Inclusion and exclusion criteria, clinical and neuroimaging protocols, and other information about C-PAS are described here[18].

**ADNI**. Seven serum metabolomics data sets from ADNI were used, including Duke2016 ($n = 818$), Duke2017 ($n = 898$), California2017 ($n = 820$), Hawaii2021 ($n = 1172$), Nightingale Health2021 ($n = 1681$), ADNI-DukeBAs2016 ($n = 833$), and ADNI-M2OVEAD2016 ($n = 897$). Demographic information and clinical data were downloaded from the ADNI data repository (www.adni-info.org and www.loni.usc.edu/ADNI/).

**ROSMAP**. The serum data set named ROSMAP-Hawaii2017 ($n = 566$) was obtained from the ROSMAP project[5,8,9]. Inclusion and exclusion criteria, clinical and pathological protocols, and other information about ROSMAP are described at https://dss.niagads.org/cohorts/religious-orders-study-memory-and-aging-project-rosmap/.

### Neuropsychological measurements and clinical diagnosis of C-PAS
Neuropsychological measurements and clinical diagnosis were performed according to our previously published reports[18]. The Chinese version of Mini-Mental State Examination (MMSE), Montreal Cognitive Assessment-Basic (MoCA-BC) and Addenbrooke's Cognitive Examination-III (ACE-III-CV) were selected as general cognitive screening tests. Different cognitive domains were assessed using a battery of standardized neuropsychological tests.

## Quantitative calculation and visual interpretation of PET images

[18 F]Florbetapir PET data were quantified using the standardized uptake value ratio (SUVr), with the cerebellum grey matter serving as the reference region. The global cortical Aβ burden was computed as the mean SUVr in cortical area, including posterior cingulate, precuneus, frontal, lateral parietal, lateral temporal, medial temporal, and occipital regions. The positive 18F-florbetapir PET images were defined through visual rating following the guidelines for interpreting amyloid PET[42]. Three physicians independently assessed all amyloid PET images, and results were determined based on a consensus, with agreement among at least two physicians. Additional details on the acquisition and preprocessing of [18 F]Florbetapir PET neuroimaging are provided in the supplementary information.

## Quantitative measurement and pretreatment of metabolic data (C-PAS)

Among numerous metabolites, 189 ones from 12 metabolite types (Fig. 2a and Table S1) that are stably measured in mammalian were quantitatively measured using UPLC-MS/MS[43]. The raw data files underwent processing using TMBQ software (V1.0, HMI, Shenzhen, China), encompassing peak integration, calibration, quantification, quality control, and batch effect adjustment for each metabolite, adhering to the manufacturer's guidelines. Outliers were identified through Cauchy distribution robust fit (K sigma=7). Any outliers ( < 0.2%) and missing values (< 0.1%) were substituted using multivariate normal imputation. To normalize their distribution for statistical analysis, the data underwent logarithmic transformation (base = 2).

## Plasma ammonia measurement

Ammonia levels in plasma samples were measured using a colorimetric assay kit (Elabscience, China) according to the manufacturer's instructions.

## Statistical analyses

Differences in clinical markers were evaluated using the Chi-squared test, student's t-test, Mann-Whitney test, analysis of variance, or Kruskal–Wallis test followed by Dunn's multiple-comparison post-hoc, as appropriate and as denoted in the text and figure legends.

Metabolites were classified into 12 types according to their chemical structure. Partial Least Squares Discriminant Analysis (PLSDA) was performed to visualize AD progression. The levels of metabolite types were represented by the first principal components (PC1s) derived from principal component analysis (PCA) based on metabolites of corresponding types. Differences in metabolite types (log transformed) were evaluated using analysis of variance test followed by Dunn's multiple-comparison post-hoc. Linear regression and logistic regression were used to identify features associated with 4 and 2 clinical stages respectively, adjusted for age, sex, BMI, education year, and APOE-ε4 status. The features were z-score scaled and the resulting effect size values were comparable. The determination of potential covariates for adjustment involved balancing the need to prevent confounding while limiting model complexity. Partial Spearman was conducted to explore associations between metabolic features and cognition, adjusted for aforementioned covariates. Voxel-wise correlations of Aβ deposition and metabolic features were analyzed by multiple variables linear regression model using SPM12 in entire cohort and sub-populations. The contributions of metabolic features to clinical practice were assessed according to the improvement of auROC derived from gradient boosting models based on basic markers alone (age, sex, education years, and APOE-ε4) and on basic markers combined with the 13 identified metabolic features. Seven-fold cross validation was performed to avoid over-fitting.

For meta-analysis, given the overlap of samples from ADNI data set (Figure S4), the three-level meta-analysis model (random effect model), a method specifically designed to address dependencies between samples or data[44], was conducted to examine the standardized mean differences of features between CN and AD groups and their associations with global cognition (partial spearman correlation adjusting age and sex). Features were scaled to 0-1 within each data set to correct batch effect. Outliers ( < 0.2%) were identified by Local Outlier Factor and were excluded from analysis[45]. Detailed information on dataset inclusion criteria, population characteristics, overlap of samples, and the three-level meta-analysis model are provided in Supplementary Information.

Data analyses were performed using R (V3.5.1), GraphPad (V9.3), and STATA (V13.0). All p-values were adjusted using the Benjamini–Hochberg's false discovery rate (FDR) and a significance level of 0.05 (two-tailed) was used unless otherwise indicated in text and figure legends. For voxel-wise analysis, the significance level was 0.05 with peak-level false discovery rate (FDR) correction and the cluster-defining voxel threshold at the default of 0.001. More details of methods are provided in supplementary method.

## Reporting summary

Further information on research design is available in the Nature Portfolio Reporting Summary linked to this article.

## Data availability

Metabolomics datasets of ADNI cohort can be accessed via the AD Knowledge Portal (https://adknowledgeportal.org; accession No. syn31513378). The full complement of clinical and demographic data for the ADNI cohort are hosted on the LONI data sharing platform and can be requested at http://adni.loni.usc.edu/data-samples/access-data/. Metabolomics datasets of C-PAS (accession No. MTBLS4554) and ROSMAP (accession No. MTBLS9583) cohorts are accessible at MetaboLights. Source data are provided with this paper.

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

## Acknowledgements

The metabolomics data of ADNI cohort were provided by the Alzheimer's Disease Metabolomics Consortium (ADMC). As such, the investigators within the ADMC, not listed specifically in this publication's author's list, provided data along with its pre-processing and prepared it for analyses, but did not participate in analyses or writing of this manuscript. A complete listing of ADMC investigators can be found at: https://sites.duke.edu/adnimetab/team/. Sample and information collection for ADNI cohorts was funded by the Alzheimer's Disease Neuroimaging Initiative (ADNI) (National Institutes of Health Grant U01 AG024904) and DOD ADNI (Department of Defense award number W81XWH-12-2-0012). ADNI is funded by the National Institute on Aging, the National Institute of Biomedical Imaging and Bioengineering, and through generous contributions from the following: AbbVie, Alzheimer's Association; Alzheimer's Drug Discovery Foundation; Araclon Biotech; BioClinica, Inc.; Biogen; Bristol-Myers Squibb Company; CereSpir, Inc.; Eisai Inc.; Elan Pharmaceuticals, Inc.; Eli Lilly and Company; EuroImmun; F. Hoffmann-La Roche Ltd and its affiliated company Genentech, Inc.; Fujirebio; GE Healthcare; IXICO Ltd.; Janssen Alzheimer Immunotherapy Research & Development, LLC.; Johnson & Johnson Pharmaceutical Research & Development LLC.; Lumosity; Lundbeck; Merck & Co., Inc.; Meso Scale Diagnostics, LLC.; NeuroRx Research; Neurotrack Technologies; Novartis Pharmaceuticals Corporation; Pfizer Inc.; Piramal Imaging; Servier; Takeda Pharmaceutical Company; and Transition Therapeutics. The Canadian

Institutes of Health Research is providing funds to support ADNI clinical sites in Canada. Private sector contributions are facilitated by the Foundation for the National Institutes of Health (https://fnih.org). The grantee organization is the Northern California Institute for Research and Education, and the study is coordinated by the Alzheimer's Disease Cooperative Study at the University of California, San Diego. Data used in preparation of this article were obtained from the Alzheimer's Disease Neuroimaging Initiative (ADNI) database (https://adni.loni.usc.edu). As such, the investigators within the ADNI contributed to the design and implementation of ADNI and/or provided data but did not participate in analyses or writing of this report. A complete listing of ADNI investigators can be found at: http://adni.loni.usc.edu/wp-content/uploads/how_to_apply/ADNI_Acknowledgement_List.pdf. This work was supported by National Key R&D Program of China (2022YFA0806400 to W.J., 2019YFA0802300 to T.C., 2021YFA1301300 to W.J., 2022ZD0213800 to F.X.), National Natural Science Foundation of China (NSFC: 31972935 to T.C., 82270917 to W.J., 82171198 to Q.G., 81974073 to W.J., and 82122012 to X.Z.).

## Author contributions

W.J. conceived the study and leaded the group. Q.G., F.P., Y.W., L.C. and Y.G. collected blood samples and scores of neuropsychological tests. F.X., J.W., Q.H. and Y.G. conducted PET test and whole brain imaging analyses. G.X., K.Z. and L.W. performed metabolome quantification. T.C., X.C., L.W., T.S., M.L., J.W., Q.H. and Y.G. conducted data pre-processing, statistical analyses, and data visualization. T.C., F.P., Q.H., and J.W. wrote the original manuscript; Q.G., F.X. and W.J. revised the manuscript. X.Z., Z.R. and A.Z. critically reviewed the manuscript and provided important intellectual content. All contributing authors have agreed to the submission of this manuscript for publication.

## Competing interests

The authors declare no competing interests.
