## [Peer Review File · Nature Communications]

Reviewers' Comments:

Reviewer #1:

Remarks to the Author:

The study focuses on the C-PAS cohort (n=1,399) comprised of a mix of individuals who were cognitively normal, had subjective cognitive complaints or who met criteria for MCI or AD. The authors quantitatively measured the abundance of 189 metabolites from non-fasting plasma. They find that conjugated bile acids, branch-chain amino acids and metabolites related to glutamate metabolism were associated with cognitive performance, clinical diagnosis, and brain amyloid deposition (in a subset of 421 individuals). They then validated these findings in the ADNI and ROSMAP cohorts (n=7,685) and found links to ammonia homeostasis which they then confirmed in blood measurements of ammonia in the C-PAS cohort. They conclude that ammonia metabolism may be important in the development of AD and may be a therapeutic target.

This is a thorough and well-designed study and a well-written manuscript. The methods are sound and well described. A major strength of the study is the large sample size of Han Chinese participants who have not been well studied in previous work. The large validation cohorts are also a strength. The metabolomic approach is somewhat limited as numerous classes of metabolites, notably many lipid classes are missing from the analysis, however the quantification of the 189 targeted metabolites is robust. The follow-up analysis of the C-PAS ammonia levels brings the study full circle from the ADNI/ROSMAP findings implicating ammonia metabolism. This is a rigorous approach. One significant weakness is the relatively young age of the C-PAS cohort (mean 66 years) which may limit the usefulness of the findings to older populations, but could be more useful for preclinical detection of AD. Another comment concerns the use of language which implies that these metabolic findings apply to AD development in an individual or group of individuals (for example lines 121-123). This study used cross-sectional data and the authors should restrict their conclusions to this methodological limitation.

Finally, the findings of a negative relationship between glutamate levels in plasma and amyloid deposition in brain beg further discussion. This finding is counter to the notion that glutamate toxicity promotes amyloid deposition and is counter to the proposed mechanism of NMDA receptor antagonists (e.g., memantine) for neuroprotection in AD. Perhaps a brief treatment of this in the Discussion section would be useful.

Reviewer #2:

Remarks to the Author:

- 1) In the introduction and throughout the authors refer to 'identified three types of metabolic features associated with ...' which is a vague reference to present findings, and should be clearly noted at first presentation within each section - that is specifically label the three feature types. For example, the heading 2.2 is "Three types of metabolites were closely associated with AD" which would be better to actually include the names of the three types in the header to highlight for the reader, or if not, then the very first sentence can state them followed by the results. This is also true in the description of Figure 2, and in the first sentence of Section 2.3.
- 2) In the final paragraph of the introduction, the authors state "We identified three types of metabolic features associated with AD stages, cognition impairment, and brain A β deposition." - is this referring to the findings of the study? Typically in the introduction the methods and hypotheses are described rather than the findings, so the terms "We identified" should be clarified. The authors also state, in this same paragraph, "Notably, these identified features are all linked to blood ammonia, a neurotoxic molecule implicated in AD pathology." Again, if this refers to the findings of the study, this should not be in the introduction but in the results and conclusions. Please clarify this final paragraph of the introduction.
- 3) In the 2.1 Study cohort, please add a brief description and/or citation on how each 'stage' was defined, as there are various approaches to defining SCD and MCI in particular.
- 4) In Section 2.2, the authors note that 'changes .. occurred early... and were associated with AD progression' but the description of the results does not clearly show the reader how the association was determined quantitatively. In addition, the text states that some metabolites 'demonstrated more pronounced alterations in multiple stages compared to CN' but again this is

not quantitatively described - what does 'more pronounced' mean in a statistical sense, or is this a qualitative comparison. Make it clear for the reader why they were elevated in importance quantitatively and qualitatively.

5) The study is cross sectional, so the authors should take care in referring to 'trajectories' and in using the terms 'increase' and 'decrease'. For example, in the end of section 2.2 statement. The use of any of these terms need to be clearly noted to refer to cross-sectional data inference, and that these changes are not being used within subjects (unless they are, and then that should be made clear). It is suggested to use greater/less than references rather than increase/decrease throughout. For example, "confirmed that ammonia levels increased gradually across clinical stages." With respect to trajectories, it is understanding that the plots are visually helpful, but both the text and the figure legends should clearly note that these are qualitative interpretations of trajectories across cross-sectional data cohorts, and that the linear/nonlinearity of the plots should not be/cannot be directly interpreted. Other cautions in this regard should be in the Discussion/conclusions, particularly if the language and use of trajectories in plots are used.

6) In the second paragraph of Section 2.3, the authors report comparisons between the CN and each other group. Differentiations between SCD and MCI would also be very informative to explore and should be included where possible.

7) In Section 2.3, second paragraph, the authors reported that SCD and AD showed BCAA-related feature alterations, why might this not have been seen in MCI?

8) In Section 2.3, brief reference is made to Figures 3, and there is a LOT of detail and information in Figure 3 in general. It is a lot for the reader to wade through, and more details should be provided to emphasize the important findings and refer to/describe the figures more carefully. In addition, how do the authors interpret any potential expectations related to these components, such as age, sex, APOE-e4, and given that these variables differ across the subject groups themselves, it should be clarified how well this was addressed in analyses and how careful/what limitations should be discussed with respect to the results.

9) In 2.4, glutamate metabolism is emphasized, but it is not clear why this specific metabolism aspect was highlighted, and the path to this should be reiterated here, as to how the significance testing directed the authors to this importance, given how many different aspects have been analyzed overall.

10) Acronym not consistent, clarify and define at earliest presentation for C-PAS (in abstract and introduction, and then in Figure 1, there is a reference to S-PAS which presumably is the same cohort?

11) Given the importance of factors such as age, please provide the range of values for age, and if relevant, the median and IQR. Although the authors note accounting for age, the overlap in age representation is critical given the tendency towards older age in the MCI and AD samples, which are smaller than the CN sample. Might the authors also consider a comparison of a more similarly age-matched CN cohort in a secondary analysis.

12) In Table 1, please provide range of age, education, and other continuous values (and IQR may also be useful); use proportions or percentages for M/F rather than counts for easier comparison across groups; clearly define all acronyms in the footnote section; clarify whether PET(%) refers to the portion of the sample that has PET data or other index? And/or proportion with positive beta amyloid?

13) Table S1 would benefit from additional text in the legend that clearly states the 12 different types used in the final column, to show the reader the types explicitly.

14) The Discussion should include a clear highlight in the first paragraph of the key take aways from the paper, from a high level, to guide the reader, given how much information is provided and the complexity of the findings. This should include both a) replication of prior findings and b) importantly, what is new and novel about the present work. Both of these categories of information should also be clear throughout the discussion section, for each finding cluster, as well as in the Conclusions.

15) In the Discussion, the authors note that "In contrast, glutamate- and BCAA-related features were lower in AD and associated with better cognition." Given the complexity of the findings, additional text should be provided to let the reader know what this exactly means - that is, for example, is having lower glutamate related features in AD associated with better cognition in AD individuals only? What does this mean?

16) Improved discussion of the impact of differences in age, education, and so on across the subgroups on the outcomes needs to be added.

Reviewer #3:

Remarks to the Author:

In this paper, Chen and colleagues perform a metabolomic analysis in a cohort of individuals from the Shanghai Preclinical Alzheimer's Disease Study (C-PAS). They first extracted 189 metabolic features from twelve different categories and used dimension reduction (PLS-DA, PCA) to identify multivariate differences between different diagnostic groups. They gradually work top-down to identify 34 specific features from the data and perform a series of linear and logistic regression analyses to determine whether there is a relationship between these features and diagnostic stage, neuropsychological assessment, and amyloid PET imaging (obtained in a subset of individuals). These features predominantly represent neurotransmitter activity (glutamate, glutamine, GABA), bile acids, and branch chain amino acids (BCAA). Substudies were performed stratifying by sex, APOE e4 status, and age. Replication was done using the ADNI and ROSMAP studies. As these features have been found to be associated with elevated blood ammonia levels, an additional standard serum test around ammonia was performed in the C-PAS cohort, which increased with disease stage and showed association with BCAA and glutamate features. The strength of this work is that multiple cohorts containing different ethnic makeups, provides more robust evidence that these markers are different at various stage of the disease. The authors speak about using these markers as an early diagnostic tool. While identifying what biological basis there is for subjective cognitive decline, what I would be even more interested in whether these markers are elevated even earlier, in individuals who are cognitively normal but show signs of AD pathology.

MAJOR COMMENTS

1. Overall: I'm having trouble understanding how the authors are placing these results in the context of what we as a field already know about the pathophysiology of the disease. Should these markers be considered a more fundamental element of AD pathology or a downstream marker of it? Are they specific to AD compared to other forms of dementia, or are they less specific to Alzheimer's disease itself and more helpful in terms of explaining heterogeneity within the disease, in terms of progression? This should be discussed a bit more in the manuscript.
2. Line 125-128: Could the authors please indicate the rationale for generating the 34 features from the 29 metabolites. What criteria (statistical testing, prior literature?) was used in terms to include these features and exclude others?
3. Section 2.4/Figure 4: For the cluster level FDR correction – the authors need to state what the cluster forming threshold was used for this work, as a liberal cluster-forming threshold can be more sensitive to false positives.
4. Section 2.4/Figure 4: The concern I have from Figure 4 is that this association is driven mostly by disease stage/severity, as most of the people with elevated amyloid PET are likely to be MCI or AD. It would be worth investigating how disease stage could be incorporated into the model, so regardless of disease severity (which has already been investigated in the other parts of the analysis), we can see what features are associated with elevated amyloid PET uptake.
5. Section 2.5: Was there any cross-validation performed in the ROC analysis to avoid overfitting?
6. Section 2.6: Multiple ADNI studies are listed as part of the meta-analysis, with a claimed total of 7,685 participants. However, there are only 2000 total participants across the various phases of ADNI. If you were to combine the sample sizes from the ADNI studies alone used in this manuscript, you would have a total of 7,119 samples, so there must be a fair amount of overlap between studies, and the question becomes if it's really surprising that similar coefficients would arise from data sets that share substantial amounts of data. The authors need to determine how many participants are overlapping between these studies and explain how they took this into account in their analysis of this data.
7. Section 2.6: It is also not clear what the inclusion criteria for this replication study is, as well as what are the differences between the different studies, particularly the ADNI ones. This would be helpful information to go in the Supplementary Material.
8. Lines 327-331: This is not sufficient enough information around the PET image processing. It states that "A consensus of physicians visually interpreted the PET images." But then there is voxelwise analysis and the use of a continuous measure in the Figures. I'm unclear where the visual reads came into play. While there is some more information in the Supplementary, it either

needs to go in the main manuscript or it needs to be referenced in the main text. There are also more details required in terms of the reference region and the target region when performing a global measure such as in Figure 5(h).

MINOR COMMENTS

1. Line 99-101: "The majority of participants were women (65.19%), and 18.51% were identified as APOE-4 positive (Table 1)." When I look at Table 1, it indicates that the positivity rate was 28.88%. Please could you explain the discrepancy?
2. Table 1: Please reduce the number of significant figures to make the data easier to read.
3. Table 1: In addition to how many participants from each group had a PET scan, the number of individuals that were classified as amyloid positive through visual read is important information to include.
4. Please replace the term gender with sex in the manuscript.
5. There is no section 4 in the manuscript.
6. Figure 2/Results: For each of the metabolite PCA's, it would be helpful to provide what percentage of variation in the data that each PC1 is capturing. This could be put into the Figure itself or somewhere in the text.
7. Figure 5(h): the x axis label is vague. It is not clear what measurement of PET is actually being performed here; it should be clear whether this is SUVR and ideally what reference region.

Reviewer #4:

Remarks to the Author:

Mass spectrometry-based metabolomics is a powerful tool for analyzing human plasma in biomarker discovery. In this manuscript, the authors employed the targeted metabolic method to analyze a total of 189 metabolites in a large-scale study encompassing over 1300 Chinese AD cohort cases. This cohort is well-documented with cognitive and neuropsychological tests, complemented by PET scans using the 18F-AV-45 tracer in 421 cases. The authors managed to correlate certain metabolite profiles with clinical data, reinforcing their findings through validation with 8 published datasets from ADNI and ROSMAP. Notably, they emphasized the discovery of elevated blood ammonia levels in AD cases. This study leverages a unique dataset and highlights potential future biomarker targets. Specific concerns are as follows:

1. The study reports a unique, extensive Chinese cohort and a wealth of clinical and metabolomic data, despite comparable studies previously carried out by ADNI and ROSMAP.
2. The finding of increased ammonia levels in AD patients has been previously reported and reviewed. It slightly diminishes the novelty of this manuscript.
3. It's imperative that all pertinent raw files, such as MS raw data and individual metabolite measurements, be made readily accessible. The current supplementary data lacks this information, even though it refers to an external website for data deposition (<http://www.metabolomicsatlas.com>). Details regarding how to access this data should be clarified.
4. The manuscript cites an Anal Chem paper for the targeted approach using UPLC-MS/MS to measure 189 metabolites. The discussion should address the pros and cons of this method and elucidate the criteria for selecting these specific metabolites to substantiate the dataset's reliability.
5. The section on data processing seems incomplete, requiring additional details on outlier removal, batch correction, and the examination of potential confounding factors.
6. To enhance the practical application of the findings, it is advisable that the authors pinpoint a small panel of metabolites to propose as promising biomarker targets.

REVIEWER COMMENTS

Reviewer #1 (Remarks to the Author):

The study focuses on the C-PAS cohort (n=1,399) comprised of a mix of individuals who were cognitively normal, had subjective cognitive complaints or who met criteria for MCI or AD. The authors quantitatively measured the abundance of 189 metabolites from non-fasting plasma. They find that conjugated bile acids, branch-chain amino acids and metabolites related to glutamate metabolism were associated with cognitive performance, clinical diagnosis, and brain amyloid deposition (in a subset of 421 individuals). They then validated these findings in the ADNI and ROSMAP cohorts (n=7,685) and found links to ammonia homeostasis which they then confirmed in blood measurements of ammonia in the C-PAS cohort. They conclude that ammonia metabolism may be important in the development of AD and may be a therapeutic target.

This is a thorough and well-designed study and a well-written manuscript. The methods are sound and well described. A major strength of the study is the large sample size of Han Chinese participants who have not been well studied in previous work. The large validation cohorts are also a strength. The metabolomic approach is somewhat limited as numerous classes of metabolites, notably many lipid classes are missing from the analysis, however the quantification of the 189 targeted metabolites is robust.

The follow-up analysis of the C-PAS ammonia levels brings the study full circle from the ADNI/ROSMAP findings implicating ammonia metabolism. This is a rigorous approach.

Answer: Thank you for your positive comments on our study design and methods used.

One significant weakness is the relatively young age of the C-PAS cohort (mean 66 years) which may limit the usefulness of the findings to older populations, but could be more useful for preclinical detection of AD.

Answer: Yes, we agree with the reviewer's comment. One notable limitation is the relatively young age of the C-PAS cohort (mean age 66), potentially constraining the generalizability of findings to older populations. However, this characteristic aligns with the cohort's original emphasis on preclinical patients, making it particularly valuable for the study of early Alzheimer's disease patients. Through participant age adjustments and stratification, we aim to augment the applicability of our results to both Alzheimer's disease and pre-Alzheimer's disease populations. This consideration has been incorporated into the discussion section, recognizing the study's limitations.

Another comment concerns the use of language which implies that these metabolic findings apply to AD development in an individual or group of individuals (for example lines 121-123). This study used cross-sectional data and the authors should restrict their conclusions to this methodological limitation.

Answer: Thank you. We have revised the manuscript carefully and corrected some inappropriate words (e.g. trajectory, increased gradually) and sentences to avoid confusion and over-interpretation.

Finally, the findings of a negative relationship between glutamate levels in plasma and amyloid deposition in brain beg further discussion. This finding is counter to the notion that glutamate toxicity promotes amyloid deposition and is counter to the proposed mechanism of NMDA receptor antagonists (e.g., memantine) for neuroprotection in AD. Perhaps a brief treatment of this in the Discussion section would be useful.

Answer: Thank you. We have addressed the inconsistency of our findings and existing notions and mechanisms on the association of glutamate and amyloid deposition in the discussion section (P11, 2nd paragraph).

Glutamate metabolism is associated with the pathological process of AD and blood levels of glutamate-related features may serve as a marker reflective of brain glutamate metabolism and A β deposition. Previous studies ^{29, 35, 36, 37} have reported contradictory results (lower levels vs higher levels) of glutamate in AD patients. Our findings of a negative relationship between glutamate levels in plasma and amyloid deposition in brain are inconsistent with the notion that elevated glutamate induces neurotoxicity, impacting neurons adversely ³⁸. Conversely, reduced glutamate levels could signify synaptic dysfunction and cognitive decline ^{39, 40}. Diverse methodologies, patient heterogeneity, and disease progression stages may collectively contribute to the observed discrepancies.

Reviewer #2 (Remarks to the Author):

1) In the introduction and throughout the authors refer to 'identified three types of metabolic features associated with ...' which is a vague reference to present findings, and should be clearly noted at first presentation within each section - that is specifically label the three feature types. For example, the heading 2.2 is "Three types of metabolites were closely associated with AD" which would be better to actually include the names of the three types in the header to highlight for the reader, or if not, then the very first sentence can state them followed by the results. This is also true in the description of Figure 2, and in the first sentence of Section 2.3.

Answer: We appreciate the reviewer's valuable input in enhancing the clarity of our presentation. We have addressed the reviewer's suggestion by explicitly naming the three feature types in the headings, figure legends, and text.

The heading 2.2 has been revised to "BAs, BCAAs, and excitatory neurotransmitters were closely associated with AD."

The description of Figure 2 now reads, "BAs, BCAAs, and excitatory neurotransmitters exhibit strong associations with AD."

The first sentence of Section 2.3 has been amended to state, "The three identified metabolite types (BAs, BCAAs, and excitatory neurotransmitters) involve 3 BCAAs, 8 excitatory neurotransmitters, and 18 bile acids."

Detailed findings in the Introduction (including the specific types of features) have been removed according to your comment 2.

2) In the final paragraph of the introduction, the authors state "We identified three types of metabolic features associated with AD stages, cognition impairment, and brain A β deposition."
- is this referring to the findings of the study? Typically in the introduction the methods and hypotheses are described rather than the findings, so the terms "We identified" should be clarified. The authors also state, in this same paragraph, "Notably, these identified features are all linked to blood ammonia, a neurotoxic molecule implicated in AD pathology." Again, if this refers to the findings of the study, this should not be in the introduction but in the results and conclusions. Please clarify this final paragraph of the introduction.

Answer: Yes, we agree with the reviewer's comment here, and revised the last paragraph of the Introduction and removed specific results and conclusions, see below.

In this study, we investigated the plasma metabolic profiles of a cohort comprising moderately aging Chinese individuals (n=1,399, mean age=66 years), encompassing both preclinical and symptomatic AD stages (Figure 1). We comprehensively identified and evaluated metabolic features associated with AD stages, cognitive impairment, and brain A β deposition within the entire cohort, as well as in stratified populations and independent cohorts. Our objective is to enhance and validate prior findings in a new cohort, contributing to an improved understanding of AD as a metabolic disorder.

3) In the 2.1 Study cohort, please add a brief description and/or citation on how each 'stage' was defined, as there are various approaches to defining SCD and MCI in particular.

Answer: Thank you. We have expanded the original descriptions on inclusion/exclusion criteria, neuropsychological measurements, and clinical diagnosis in SI, see below. A reference (PMID: 37357299) on the C-PAS cohort (which include the definition of clinical stages) has been cited in the text.

SI Method-Neuropsychological measurements and clinical diagnosis in C-PAS

All participants in the C-PAS study had to fulfill the following criteria: (1) Age between 40 and 85 years with a minimum education duration of more than 1 year; (2) Absence of severe hearing or visual impairment, and proficiency in Mandarin communication; (3) Willingness to complete neuropsychological tests, cranial MRI, brain PET scans, and blood biomarker assessments; (4) No history of stroke, craniocerebral injury, brain tumor, anxiety, depression, or other conditions potentially impacting cognitive function adversely.

Brief cognitive screening tests included the Chinese versions of Mini-Mental State Examination (MMSE)¹, Montreal Cognitive Assessment-Basic (MoCA-BC)², and Addenbrooke's Cognitive Examination-III (ACE-III-CV)³. Various cognitive domains were evaluated through standardized neuropsychological tests: Auditory Verbal Learning Test (AVLT)⁴ and Brief Visuospatial Memory Test-Revised (BVMT-R)⁵ for memory; Boston Naming Test (BNT)⁶ and Animal Verbal Fluency Test (AFT)⁷ for language; Shape Trail Test Part A and B (STT-A, STT-B)⁸ for executive function; Judgement of Line Orientation (JLO)⁹ for visuospatial ability; Symbol Digit Modalities Test (SDMT) and Digit Span Test (DST)¹⁰ for attention. Global functional status was assessed using Activities of Daily Living (ADL)¹¹ and Functional Assessment Questionnaire (FAQ)¹².

Participants with no cognitive complaint and objective cognitive impairment assessed via neuropsychological tests were defined as cognitively normal (CN). Those with self-reported memory decline but performed essentially normal on neuropsychological tests were classified as Subjective Cognitive Decline (SCD) according to the conceptual framework proposed by the working group of SCD Initiative (SCD-I)¹³. Mild cognitive impairment (MCI) was diagnosed according to the actuarial neuropsychological criteria put forward by Jak and Bondi¹⁴. Participants were classified as having dementia according to the criteria of Diagnostic and Statistical Manual of Mental Disorders, 4th edition- revised, and the clinical diagnosis of probable AD dementia was made according to the NIA-AA criteria¹⁵.

4) In Section 2.2, the authors note that ‘changes .. occurred early... and were associated with AD progression’ but the description of the results does not clearly show the reader how the association was determined quantitatively. In addition, the text states that some metabolites ‘demonstrated more pronounced alterations in multiple stages compared to CN’ but again this is not quantitatively described - what does ‘more pronounced’ mean in a statistical sense, or is this a qualitative comparison. Make it clear for the reader why they were elevated in importance quantitatively and qualitatively.

Answer : Thank you. We have revised this section accordingly and clarified the changes and the associations.

“Partial least squares discriminant analysis (PLS-DA) was employed to analyze the concentrations of all metabolites. The centroid scores plot illustrated distinct alterations in metabolic profiles at various disease stages, with the severity of the disease corresponding to an increased distance from the cognitively normal (CN) profile (Figure 2b).”

“In comparison to CN, the levels of three marker types, BAs, BCAAs, and excitatory neurotransmitters, exhibited significant alterations (post hoc Dunnett's test $p < 0.05$) across more disease stages than other metabolites (Figure 2c).”

5) The study is cross sectional, so the authors should take care in referring to ‘trajectories’ and

in using the terms 'increase' and 'decrease'. For example, in the end of section 2.2 statement. The use of any of these terms need to be clearly noted to refer to cross-sectional data inference, and that these changes are not being used within subjects (unless they are, and then that should be made clear). It is suggested to use greater/less than references rather than increase/decrease throughout. For example, "confirmed that ammonia levels increased gradually across clinical stages." With respect to trajectories, it is understanding that the plots are visually helpful, but both the text and the figure legends should clearly note that these are qualitative interpretations of trajectories across cross-sectional data cohorts, and that the linear/nonlinearity of the plots should not be/cannot be directly interpreted. Other cautions in this regard should be in the Discussion/conclusions, particularly if the language and use of trajectories in plots are used.

Answer: We appreciate the reviewer's insightful comments regarding the cross-sectional nature of our study. In response, we have diligently revised the manuscript to avoid inappropriate terms such as 'trajectories,' 'increase,' 'decrease,' and 'decline,' which could be potentially misleading. Throughout the manuscript, we have replaced these terms with greater/less than references to accurately reflect the cross-sectional data.

For example, the last sentence of section 2.2 has been adjusted to: "We further observed the levels of five sub-types belonging to these three types in 4 clinical stages and found that the level of conjugated BAs was the highest in AD and the lowest in CN (AD>MCI>SCD>CN), and the level of glutamate-related metabolites was the lowest in AD and the highest in CN (AD<MCI<SCD<AD; Figure 2d)."

The results of ammonia data analysis (2.7) have been modified to: "...confirmed that ammonia levels were the highest in AD and the lowest in CN (AD>MCI>SCD>CN; Figure 7b)."

The legend of Figure 2d has been revised to: "d) Levels of five sub-types belonging to the top three types of Figure 2c in four clinical stages. The levels are represented by the PC1 scores derived from PCA based on metabolites belonging to each sub-type and are fitted by LOESS."

6) In the second paragraph of Section 2.3, the authors report comparisons between the CN and each other group. Differentiations between SCD and MCI would also be very informative to explore and should be included where possible.

Answer: We have conducted additional analyses, and the distinctions (including effect sizes and p-values) between each pair of stages are illustrated in Figure 3g. The relevant text has been revised accordingly, as addressed in the response to your question 8. Notably, we observed that the number of differentially significant ($p < 0.05$) features when discriminating between SCD and MCI (M4) was smaller than in other comparisons. In women, *APOE-ε4* non-carriers, and individuals under 65 years old, three BCAA-related features, three glutamate-related features, and one BCAA and one BA were found to differ between SCD and MCI respectively. Conversely, in men, *APOE-ε4* carriers, and individuals over 65 years old, none of the 13 features exhibited differences between these two stages.

Figure 3. g) Effect sizes of 13 metabolic features when differentiating CN and SCD (M1), CN and MCI (M2), CN and AD (M3), SCD and MCI (M4), SCD and AD (M5), and MCI and AD (M6) respectively in stratified populations. Colored cell indicates logistic regression $p < 0.05$ and * indicates $p < 0.01$. Covariates including age, sex, BMI, education year, and *APOE-ε4* were adjusted.

7) In Section 2.3, second paragraph, the authors reported that SCD and AD showed BCAA-related feature alterations, why might this not have been seen in MCI?

Answer: Thank you for the insightful question; we have duly considered this aspect. Our speculation revolves around the possibility that BCAAs may undergo nonlinear changes in tandem with the onset and progression of AD. Consequently, the alterations observed in SCD and AD, compared to CN, were more pronounced, while those in MCI did not attain statistical significance. Additionally, it's plausible that the association between BCAAs and AD might be confounded by various factors, given that BCAAs are acknowledged risk factors for insulin resistance, diabetes, and cardiovascular events—themselves independent risk factors for AD. As acknowledged in our Discussion (P11, line 1-4), the existing evidence is insufficient to fully elucidate the impact of BCAAs on AD progression. Ongoing investigations, incorporating longitudinal designs and accounting for factors like nonlinear changes, complications, medications, and diet, are underway.

8) In Section 2.3, brief reference is made to Figures 3, and there is a LOT of detail and information in Figure 3 in general. It is a lot for the reader to wade through, and more details should be provided to emphasize the important findings and refer to/describe the figures more carefully. In addition, how do the authors interpret any potential expectations related to these components, such as age, sex, *APOE-ε4*, and given that these variables differ across the subject groups themselves, it should be clarified how well this was addressed in analyses and how careful/what limitations should be discussed with respect to the results.

Answer: We have carefully revised section 2.3 according to the reviewer's suggestion. Figure 3 was rearranged and expanded. Additional descriptions on the consideration of covariates and other findings were provided. The discussion section was also revised. Due to the word limit,

more discussions were provided in SI-discussion to address the potential impact of age, sex, and *APOE-ε4* status on the association of metabolic profiles and AD progression (see the answer to your question 16).

Section 2.3 after revision:

2.3 Conjugated BAs, BCAAs, and Glutamate-Related Features: Stage-Specific and Population-Specific Associations

..... We constructed linear regression models (age, sex, BMI, *APOE-ε4*, and education year were adjusted) and identified 13 features (out of 63 comprising 29 metabolites and 34 extended features) significantly associated ($FDR < 0.05$) with clinical stages within the entire population (refer to the first column in Figure 3a). Notably, six features related to BAs (five of which were linked to conjugated BAs), four to BCAAs, and three to excitatory neurotransmitters (all involving glutamate) exhibited associations. Positive associations (with positive effect sizes) were observed for BA-related features, whereas BCAA and glutamate-related features displayed negative associations (with negative effect sizes) in tandem with disease severity. Subsequently, logistic regression models were constructed on these 13 features, facilitating the differentiation of every two stages (M1-M6) while adjusting for covariates. Interestingly, the number of features with different levels ($p < 0.05$) was smaller in SCD and MCI (M4), while CN and AD (M3) exhibited a larger disparity than other comparisons within the entire population (Figure 3a). The stage-specific and complementary performances of these features were evident, as distinctions were observed in BA-related and glutamate-related features across all comparisons, except for M4 (SCD vs. MCI) and M1 (CN vs. CSD) respectively while BCAA-related features showed differences between M1 (CN vs. SCD) and M3 (CN vs. AD). Furthermore, the association patterns of these features with cognition were evaluated (Figure 3b). Comparatively, glutamate-related features exhibited the highest number of associations, followed by BA- and BCAA-related features.

Stratification analysis, accounting for age, sex, and *APOE-ε4* characteristics, revealed diverse patterns across sub-populations (Figures 3c-3e). *APOE-ε4* status exerted a more substantial impact on these features compared to sex and age. A lower number of differential features was observed among four stages (Figure 3f) and between every two stages (Figure 3g), as well as in features associated with cognition (Figure S3) among *APOE-ε4* carriers compared to non-carriers. Men exhibited a higher number of BA-related features than women across four stages (Figure 3f), between every two stages (Figure 3g), and in associations with cognition (Figure S3). Conversely, women and younger participants showed a greater prevalence of BCAA-related features in these scenarios.

The related sentences in discussion after revision:

“... We observed that their performances were not only stage-specific but also population-specific. This implies that employing distinct sets of metabolic features tailored to specific

stages and populations enhances the precision in characterizing the metabolic patterns of AD. On the other hand, caution is warranted in interpreting our findings. Further analysis, incorporating additional covariates and exploring their interactions, will contribute to a more nuanced understanding of these features and the overall metabolic landscape in AD. More in-depth discussions on the potential influences of age, sex, and *APOE*- ϵ 4 status on the association between metabolic profiles and the progression of AD are provided in supplementary information."

Figure 3. Conjugated BAs, BCAAs, and glutamate-related features were associated with clinical stages and cognition in a stage-specific and population-specific way. a) Effect sizes of 13 metabolic features significantly different among four clinical stages (M0) and their performances in differentiating every two stages (M1-M6) based on linear regression models (M0) and logistic regression models (M1-M6) respectively. Data from entire population was used. Colored cell

indicates $FDR < 0.05$ (M0) or $p < 0.05$ (M1-M6) and * indicates $FDR < 0.01$ (M0) or $p < 0.01$ (M1-M6). M0: CN-SCD-MCI-AD; M1: SCD vs. CN; M2: CN vs. MCI; M3: CN vs. AD; M4: SCD vs. MCI; M5: SCD vs. AD; M6: MCI vs. AD. b) Associations of 13 metabolic features and cognition scores (entire population). Cell color indicates correlation coefficient from Partial Spearman analysis (red: positive; blue: negative; blank: $p \geq 0.05$). c) The distribution of men and women in four clinical stages. d) The distribution of *APOE-ε4* carriers and non-carriers in four clinical stages. e) Age distribution of participants in four clinical stages. f) Effect sizes of 13 metabolic features when differentiating four clinical stages based on linear regression models (M0) in stratified populations. Colored cell indicates $FDR < 0.05$ and * indicates $FDR < 0.01$. g) Effect sizes of 13 metabolic features when differentiating CN and SCD (M1), CN and MCI (M2), CN and AD (M3), SCD and MCI (M4), SCD and AD (M5), and MCI and AD (M6) respectively in stratified populations. Colored cell indicates $p < 0.05$ and * indicates $p < 0.01$. The correlation coefficients of 13 metabolic features and cognition scores based on Partial Spearman in stratified populations are shown in Figure S2. Covariates including age, sex, BMI, education year, and *APOE-ε4* were adjusted in all the analyses.

GCDCA: Chenodeoxycholic acid glycine conjugate; GDCA: Deoxycholic acid glycine conjugate; GCDCA/CDCA: the ratio of Chenodeoxycholic acid glycine conjugate and Chenodeoxycholic acid; GDCA/DCA: the ratio of Deoxycholic acid glycine conjugate and Deoxycholic acid; GDCA/CA: the ratio of Deoxycholic acid glycine conjugate and Cholic acid; PriBA: concentration summation of primary BAs (CA+CDCA+GCA+TCA+GCDCA+TCDCA); MMSE: Mini-Mental State Examination; ACEIII-CV: Chinese version of Addenbrooke's cognitive examination-III; MoCA-BC: Chinese version of Montreal Cognitive Assessment-Basic; FAQ: Functional Assessment Questionnaire; ADL: Activities of Daily Living; AVLT: Auditory Verbal Learning Test; BVMT-R: Brief Visuospatial Memory Test-Revised; N4: short delayed recall; N5: long delayed recall; N6/N7: recognition; AFT: Animal Verbal Fluency Test; BNT: Boston Naming Test; STT-A and B: Shape Trail Test Part A and B; JLO: Judgement of Line Orientation; SDMT: Symbol Digit Modalities Test; DST: Digit Span Test.

9) In 2.4, glutamate metabolism is emphasized, but it is not clear why this specific metabolism aspect was highlighted, and the path to this should be reiterated here, as to how the significance testing directed the authors to this importance, given how many different aspects have been analyzed overall.

Answer: Statistical analysis was undertaken to assess the associations of the 13 identified features with brain $A\beta$ deposition. In addition to presenting the outcomes of whole-brain association analysis (refer to Figures 4d-4n), we compared the levels of these 13 features between participants with positive and negative brain $A\beta$ deposition, determined by the consensus of physicians' visual interpretation of PET images (refer to Figures 4a-4c). Our findings revealed that glutamate-related features exhibited superior performance compared to others, demonstrating more significant associations with brain $A\beta$ deposition and consistent

differences between A β + and A β - individuals, both across the entire cohort and within stratified subgroups. A comprehensive revision of Section 2.4 has been undertaken to provide clarity on this matter (please refer to the response to Reviewer 3, Question 1).

10) Acronym not consistent, clarify and define at earliest presentation for C-PAS (in abstract and introduction, and then in Figure 1, there is a reference to S-PAS which presumably is the same cohort?

Answer: Thank you for pointing it out. Yes, they are the same cohort and “S-PAS” in figure 1 and SI has been corrected to C-PAS.

11) Given the importance of factors such as age, please provide the range of values for age, and if relevant, the median and IQR. Although the authors note accounting for age, the overlap in age representation is critical given the tendency towards older age in the MCI and AD samples, which are smaller than the CN sample. Might the authors also consider a comparison of a more similarly age-matched CN cohort in a secondary analysis.

Answer: Thank you. In response to the reviewer's comment, Table 1 has been revised to include the range, median, and interquartile range (IQR) of age. Additionally, we conducted a secondary analysis using age-matched participants (Table S3), and the key performances of the 13 identified features were found to be consistent with those of the full population (refer to Figure S3). A brief description has been added in Section 2.6.

A new paragraph has been included in Section 2.6:

Given the importance of age, we examined the performance of the 13 features in an age-matched sub-population (Table S3, n=991, mean age=69.7, ranging from 60 to 89), despite age adjustments being made in the above analyses. In line with the entire population, BA-related features showed positive associations, while BCAA and glutamate-related features exhibited negative associations with disease severity (Figure S3a). Glutamate-related features displayed the highest number of associations with cognition, followed by BA- and BCAA-related features (Figure S3b). Sex and APOE- ϵ 4 stratified analyses indicated that APOE- ϵ 4 status had a more substantial impact than sex (Figures S3c and S3d).

Newly added Table S3:

Table S3. Characteristics of age-matched sub-population.

Characteristic	ALL (n=991)	CN (n=259)	SCD (n=141)	MCI(n=225)	AD (n=366)
Age (yr)	69.7+6.5	69.6+4.6	69.4+4.7	69.5+6.0	69.9+8.4
[min, max]	[60,89]	[64,84]	[63,81]	[60,86]	[60,89]
Median (IQR)	69(65,74)	69(66,73)	69(65,72)	69(65,74)	70(65,76)
Sex (Men%)	37.8%	40.2%	35.5%	35.7%	38.4%

BMI (kg/m²)	23.2+3.3*	23.6+3.2	23.5+3.1	23.1+3.4	22.9+3.4#
[min, max]	[13.7,33.8]	[15.4,33.8]	[16.4,30.2]	[15.5,33.2]	[13.7,31.1]
Median (IQR)	23.1(21.0,25.3)	23.5(21.5,25.3)	23.5(21.2,25.9)	23.0(20.8,25.2)	22.7(20.7,25.1)
Education(yr)	11.0+3.2*	11.8+3.1	11.6+3.2	11.1+3.0#	10.1+3.2#
[min, max]	[6, 22]	[6, 20]	[6,18]	[6, 22]	[6, 19]
Median (IQR)	11(9,13)	12(9,14)	12(9,14)	11(9,13)	10(7.2,12)
APOE (ε4) carrier %^a	31.9%*	17.4%	17.7%	31.6%#	47.8%#
Brain AβPET (%)^b	28.3%*	36.3%	39.7%	28.4%	18.0%#
Brain Aβ+(%) ^c	26.1%	14.9%	17.8%	21.8%	53.0%
MMSE	23.4+6.0*	28.0+1.7	27.4+1.8#	26.3+2.0#	16.9+4.7#
[min, max]	[10,30]	[21,30]	[21,30]	[15,30]	[10,27]
Median (IQR)	26(20,28)	28(27,29)	28(26,29)	27(25,28)	18(12,21)
ACEIII-CV	64.9+18.7*	80.8+7.9	77.1+7.7#	69.8+8.4#	45.6+14.7#
[min, max]	[10,97]	[60,97]	[60,95]	[50,94]	[10,77]
Median (IQR)	69(54,79)	81.5(76,87)	77(72,82)	71(64,75.2)	48(36,58)
MoCA-BC	22.1+4.9*	25.6+2.6	24.0+3.0#	21.7+3.3#	15.3+3.3#
[min, max]	[10,30]	[20,30]	[17,29]	[15,30]	[10,22]
Median (IQR)	23(19,26)	26(24,27)	24(22,27)	22(20,24)	15(13,18)

Data are presented as mean+S.D., range, median with interquartile range (IQR), or percentage.

* indicates Chi-squared test, analysis of variance, or Kruskal–Wallis test FDR<0.05 when comparing 4 groups (adjusted by Benjamini and Hochberg). # indicates Chi-squared test, student's t-test or Mann-Whitney test FDR<0.05 when compared to CN (adjusted by Benjamini and Hochberg). CN: cognitively normal; AD: Alzheimer's disease; SCD: subjective cognitive decline; MMSE: Mini-Mental State Examination; ACEIII-CV: Chinese version of Addenbrooke's cognitive examination-III; MoCA-BC: Chinese version of Montreal Cognitive Assessment-Basic. a: the percentage of APOE-ε4 carriers. b: the percentage of the participants that accepted brain PET test. c: the percentage of participants with positive Aβ in those accepted brain AV45-PET test.

Newly added Figure S3:

Figure S3. Performances of the 13 identified features in age-matched sub-population. a) Effect sizes of the features among four clinical stages (M0) and between every two stages (M1-M6) based on linear regression models (M0) and logistic regression models (M1-M6) respectively. Colored cell indicates $p < 0.05$ and * indicates $p < 0.01$. b) Associations of the features and cognition scores. Cell color indicates correlation coefficient from Partial Spearman analysis. c) Effect sizes of the features when differentiating four clinical stages based on linear regression models (M0) in sex and APOE-ε4 stratified populations. Colored cell indicates $p < 0.05$ and * indicates $p < 0.01$. d) Effect sizes of the features when differentiating every two stages in sex and APOE-ε4 stratified populations. Colored cell indicates $p < 0.05$ and * indicates $p < 0.01$. Covariates including age, sex, BMI, education year, and APOE-ε4 were adjusted in all the analysis when applicable.

12) In Table 1, please provide range of age, education, and other continuous values (and IQR may also be useful); use proportions or percentages for M/F rather than counts for easier comparison across groups; clearly define all acronyms in the footnote section; clarify whether PET(%) refers to the portion of the sample that has PET data or other index? And/or proportion with positive beta amyloid?

Thank you. Table 1 has undergone revision, now incorporating the range, median, and interquartile range (IQR) of continuous values, such as age, BMI, education year, MMSE, ACEIII-CV, and MoCA-BC. Percentages, rather than counts, have been presented for the male/female distribution to facilitate easier cross-group comparisons. Additionally, all acronyms

have been clearly defined in the footnote section. Regarding PET(%), it refers to the proportion of the sample that underwent a brain PET test. The percentage of participants with positive A β in those who underwent the brain AV45-PET test has also been included.

The revised Table 1:

Table 1. Characteristics of study population from C-PAS cohort

Characteristic	ALL (n=1399)	CN(n=489)	SCD(n=239)	MCI(n=284)	AD(n=387)
Age (yr)	66.1+8.6*	63.5+8.2	64.4+7.4	66.5+8.1#	70.4+8.4#
[min, max]	[31, 89]	[31, 84]	[47, 81]	[43, 86]	[41, 89]
Median (IQR)	66 (60, 72)	64 (58, 69)	64 (58, 70)	66 (61, 73)	71 (65, 77)
Sex (Men%)	34.4%	33.5%	28.9%	34.5%	38.8%
BMI (kg/m ²)	23.5+3.8	23.5+3.4	24.0+3.5	23.1+3.3	22.9+3.4
[min, max]	[13.7, 33.8]	[15.4, 33.8]	[16.4, 31.6]	[15.5, 33.2]	[13.7, 31.1]
Median (IQR)	23.2 (21.0, 25.4)	23.3 (21.2, 25.3)	23.7 (21.5, 26.0)	23.0 (20.9, 25.4)	22.7 (20.6, 25.2)
Education (yr)	11.4+3.2*	12.4+3.1	11.9+3.1	11.1+3.0	10.2+3.2#
[min, max]	[6, 22]	[6, 22]	[6, 20]	[6, 22]	[6, 19]
Median (IQR)	11 (9, 14)	12 (10, 15)	12 (9, 14)	11 (9, 12)	10 (7, 12)
APOE (ϵ 4) carrier % ^a	28.9%*	19.2%	17.6%	29.6%#	47.6%#
Brain A β PET(%) ^b	30.1%*	34.8%	39.3%	31.7%#	17.3%#
Brain A β +(%) ^c	22.1%	14.1%	17.0%	20.0%	52.2%#
MMSE	24.6+5.6*	28.2+1.7	27.7+1.8#	26.5+2.1#	16.8+4.7#
[min, max]	[10, 30]	[20, 30]	[21, 30]	[15, 30]	[10, 27]
Median (IQR)	27 (22, 29)	28.5 (27, 29)	28 (26, 29)	27 (25, 28)	17.5 (12, 21)
ACEIII-CV	68.8+18.2*	82.0+7.9	77.8+8.0#	70.3+9.0#	45.7+14.7#
[min, max]	[10, 97]	[60, 97]	[60, 96]	[50, 94]	[10, 77]
Median (IQR)	73 (60, 82)	83 (77, 88)	78 (73, 83)	71 (64, 76)	48 (36, 58)
MoCA-BC	23.3+4.8*	26.1+2.5	24.7+3.0#	21.9+3.4#	15.2+3.3#

[min, max]	[10, 30]	[20, 30]	[17, 30]	[15, 30]	[10, 22]
Median (IQR)	24.50 (20, 27)	27 (25, 28)	25 (22, 27)	22 (20, 24)	15 (12, 18)

Data are presented as mean+S.D., range, median with interquartile range (IQR), or percentage.

* indicates Chi-squared test, analysis of variance, or Kruskal–Wallis test FDR<0.05 when comparing 4 groups (adjusted by Benjamini and Hochberg). # indicates Chi-squared test, student's t-test or Mann-Whitney test FDR<0.05 when compared to CN (adjusted by Benjamini and Hochberg). CN: cognitively normal; AD: Alzheimer's disease; SCD: subjective cognitive decline; MMSE: Mini-Mental State Examination; ACEIII-CV: Chinese version of Addenbrooke's cognitive examination-III; MoCA-BC: Chinese version of Montreal Cognitive Assessment-Basic. a: the percentage of APOE-ε4 carriers. b: the percentage of the participants that accepted brain PET test. c: the percentage of participants with positive Aβ in those who underwent the brain AV45-PET scans.

13) Table S1 would benefit from additional text in the legend that clearly states the 12 different types used in the final column, to show the reader the types explicitly.

Answer: Thank you. We have provided the description of the 12 types in the final column of Table S1.

Table S1. Quantified metabolites

index	name	HMDB	type
1	1-Methylhistidine	HMDB0000001	Amino Acids
2	Beta-Alanine	HMDB0000056	Amino Acids
3	Creatine	HMDB0000064	Amino Acids
4	L-Tyrosine	HMDB0000158	Amino Acids
5	L-Phenylalanine	HMDB0000159	Amino Acids
6	L-Alanine	HMDB0000161	Amino Acids
7	L-Proline	HMDB0000162	Amino Acids
8	L-threonine	HMDB0000167	Amino Acids
9	L-Asparagine	HMDB0000168	Amino Acids
10	L-Histidine	HMDB0000177	Amino Acids
11	L-Lysine	HMDB0000182	Amino Acids
12	L-Serine	HMDB0000187	Amino Acids
13	Ornithine	HMDB0000214	Amino Acids
14	Sarcosine	HMDB0000271	Amino Acids
15	L-Arginine	HMDB0000517	Amino Acids
16	glutamine	HMDB0000641	Amino Acids
17	Homocitrulline	HMDB0000679	Amino Acids

18	L-Methionine	HMDB0000696	Amino Acids
19	L-Pipecolic acid	HMDB0000716	Amino Acids
20	4-Hydroxyproline	HMDB0000725	Amino Acids
21	Citrulline	HMDB0000904	Amino Acids
22	L-Tryptophan	HMDB0000929	Amino Acids
23	5-Aminolevulinic acid	HMDB0001149	Amino Acids
24	Methylcysteine	HMDB0002108	Amino Acids
25	N-Acetylserine	HMDB0002931	Amino Acids
26	3-Aminoisobutanoic acid	HMDB0003911	Amino Acids
27	Phenylacetylglutamine	HMDB0006344	Amino Acids
28	L-Isoleucine	HMDB0000172	BCAAs
29	L-Leucine	HMDB0000687	BCAAs
30	L-Valine	HMDB0000883	BCAAs
31	glutamate	HMDB0000148	excitatory neurotransmitters
32	L-Aspartic acid	HMDB0000191	excitatory neurotransmitters
33	Pyroglutamic acid	HMDB0000267	excitatory neurotransmitters
34	N-Acetyl-L-aspartic acid	HMDB0000812	excitatory neurotransmitters
35	Dimethylglycine	HMDB0000092	inhibitory neurotransmitters
36	GABA	HMDB0000112	inhibitory neurotransmitters
37	Glycine	HMDB0000123	inhibitory neurotransmitters
38	Acetylglycine	HMDB0000532	inhibitory neurotransmitters
39	2-Phenylglycine	HMDB0002210	inhibitory neurotransmitters
40	TCA	HMDB0000036	Bile Acids
41	GCA	HMDB0000138	Bile Acids
42	7_DHCA	HMDB0000391	Bile Acids
43	CDCA	HMDB0000518	Bile Acids
44	CA	HMDB0000619	Bile Acids
45	DCA	HMDB0000626	Bile Acids
46	GDCA	HMDB0000631	Bile Acids
47	GCDCA	HMDB0000637	Bile Acids
48	isoDCA	HMDB0000686	Bile Acids
49	GLCA	HMDB0000698	Bile Acids
50	GUDCA	HMDB0000708	Bile Acids
51	HCA	HMDB0000760	Bile Acids
52	LCA	HMDB0000761	Bile Acids
53	TUDCA	HMDB0000874	Bile Acids
54	TDCA	HMDB0000896	Bile Acids
55	UDCA	HMDB0000946	Bile Acids

56	TCDCA	HMDB0000951	Bile Acids
57	GHCA	HMDB0240607	Bile Acids
58	NorCA	Norcholic acid	Bile Acids
59	Glyceric acid	HMDB0000139	Carbohydrates
60	Gluconolactone	HMDB0000150	Carbohydrates
61	N-Acetylneuraminic acid	HMDB0000230	Carbohydrates
62	Galactonic acid	HMDB0000565	Carbohydrates
63	Erythronic acid	HMDB0000613	Carbohydrates
64	Glucaric acid	HMDB0000663	Carbohydrates
65	Ribonic acid	HMDB0000867	Carbohydrates
66	Threonic acid	HMDB0000943	Carbohydrates
67	Tartaric acid	HMDB0000956	Carbohydrates
68	D-Xylose	HMDB0000098	sugars
69	D-Glucose	HMDB0000122	sugars
70	D-Maltose	HMDB0000163	sugars
71	D-Fructose	HMDB0000660	sugars
72	Rhamnose	HMDB0000849	sugars
73	D-Xylulose	HMDB0001644	sugars
74	L-Carnitine	HMDB0000062	Carnitines
75	L-Acetylcarnitine	HMDB0000201	Carnitines
76	Palmitoylcarnitine	HMDB0000222	Carnitines
77	2-Methylbutyroylcarnitine	HMDB0000378	Carnitines
78	Decanoylcarnitine	HMDB0000651	Carnitines
79	Isovalerylcarnitine	HMDB0000688	Carnitines
80	Hexanoylcarnitine	HMDB0000756	Carnitines
81	Octanoylcarnitine	HMDB0000791	Carnitines
82	Propionylcarnitine	HMDB0000824	Carnitines
83	Malonylcarnitine	HMDB0002095	Carnitines
84	Dodecanoylcarnitine	HMDB0002250	Carnitines
85	Oleoylcarnitine	HMDB0005065	Carnitines
86	Tetradecanoylcarnitine	HMDB0005066	Carnitines
87	Linoleyl carnitine	HMDB0006469	Carnitines
88	Valerylcarnitine	HMDB0013128	Carnitines
89	Glutaryl carnitine	HMDB0013130	Carnitines
90	Methylmalonylcarnitine	HMDB0013133	Carnitines
91	3-Hydroxyisovalerylcarnitine	HMDB0061189	Carnitines
92	O-Adipoylcarnitine	HMDB0061677	Carnitines

93	4-Methylhexanoic acid	4-Methylhexanoic acid	Fatty Acids
94	Oleic acid	HMDB0000207	Fatty Acids
95	2-Hydroxy-3-methylbutyric acid	HMDB0000407	Fatty Acids
96	Citramalic acid	HMDB0000426	Fatty Acids
97	Adipic acid	HMDB0000448	Fatty Acids
98	Capric acid	HMDB0000511	Fatty Acids
99	5Z-Dodecenoic acid	HMDB0000529	Fatty Acids
100	3-Methyladipic acid	HMDB0000555	Fatty Acids
101	Dodecanoic acid	HMDB0000638	Fatty Acids
102	Heptanoic acid	HMDB0000666	Fatty Acids
103	Linoleic acid	HMDB0000673	Fatty Acids
104	Azelaic acid	HMDB0000784	Fatty Acids
105	Sebacic acid	HMDB0000792	Fatty Acids
106	Myristic acid	HMDB0000806	Fatty Acids
107	Pimelic acid	HMDB0000857	Fatty Acids
108	Suberic acid	HMDB0000893	Fatty Acids
109	Tridecanoic acid	HMDB0000910	Fatty Acids
110	Undecanoic acid	HMDB0000947	Fatty Acids
111	Arachidonic acid	HMDB0001043	Fatty Acids
112	Alpha-Linolenic acid	HMDB0001388	Fatty Acids
113	2-Hydroxycaproic acid	HMDB0001624	Fatty Acids
114	Methylsuccinic acid	HMDB0001844	Fatty Acids
115	Docosapentaenoic acid (22n-6)	HMDB0001976	Fatty Acids
116	2-Hydroxy-2-methylbutyric acid	HMDB0001987	Fatty Acids
117	Eicosapentaenoic acid	HMDB0001999	Fatty Acids
118	Myristoleic acid	HMDB0002000	Fatty Acids
119	2,2-Dimethylsuccinic acid	HMDB0002074	Fatty Acids
120	Docosahexaenoic acid	HMDB0002183	Fatty Acids
121	Adrenic acid	HMDB0002226	Fatty Acids
122	Dihomo-gamma-linolenic acid	HMDB0002925	Fatty Acids
123	Gamma-Linolenic acid	HMDB0003073	Fatty Acids
124	Palmitoleic acid	HMDB0003229	Fatty Acids
125	Docosapentaenoic acid (22n-3)	HMDB0006528	Fatty Acids

126	Ricinoleic acid	HMDB0034297	Fatty Acids
127	12-hydroxystearic acid	HMDB0061706	Fatty Acids
128	9E-tetradecenoic acid	HMDB0062248	Fatty Acids
129	Caprylic acid	HMDB0000482	Fatty Acids
130	2-Hydroxybutyric acid	HMDB0000008	Organic Acids
131	Alpha-ketoisovaleric acid	HMDB0000019	Organic Acids
132	cis-Aconitic acid	HMDB0000072	Organic Acids
133	Citric acid	HMDB0000094	Organic Acids
134	Glycolic acid	HMDB0000115	Organic Acids
135	Guanidoacetic acid	HMDB0000128	Organic Acids
136	Fumaric acid	HMDB0000134	Organic Acids
137	L-Malic acid	HMDB0000156	Organic Acids
138	Maleic acid	HMDB0000176	Organic Acids
139	L-Lactic acid	HMDB0000190	Organic Acids
140	Isocitric acid	HMDB0000193	Organic Acids
141	Methylmalonic acid	HMDB0000202	Organic Acids
142	Oxoglutaric acid	HMDB0000208	Organic Acids
143	Oxoadipic acid	HMDB0000225	Organic Acids
144	Pyruvic acid	HMDB0000243	Organic Acids
145	Succinic acid	HMDB0000254	Organic Acids
146	3-Hydroxybutyric acid	HMDB0000357	Organic Acids
147	3-Methyl-2-oxovaleric acid	HMDB0000491	Organic Acids
148	D-2-Hydroxyglutaric acid	HMDB0000606	Organic Acids
149	Glutaconic acid	HMDB0000620	Organic Acids
150	Glutaric acid	HMDB0000661	Organic Acids
151	Malonic acid	HMDB0000691	Organic Acids
152	Hydroxypropionic acid	HMDB0000700	Organic Acids
153	Alpha-Hydroxyisobutyric acid	HMDB0000729	Organic Acids
154	Benzoic acid	HMDB0001870	Organic Acids
155	Oxalic acid	HMDB0002329	Organic Acids
156	Quinic acid	HMDB0003072	Organic Acids
157	Ketoleucine	HMDB0000695	Organic Acids
158	p-Hydroxyphenylacetic acid	HMDB0000020	others
159	Homovanillic acid	HMDB0000118	others
160	Indoleacetic acid	HMDB0000197	others
161	Phenylpyruvic acid	HMDB0000205	others

162	Phenylacetic acid	HMDB0000209	others
163	Ortho-Hydroxyphenylacetic acid	HMDB0000669	others
164	Indolelactic acid	HMDB0000671	others
165	Mandelic acid	HMDB0000703	others
166	Hippuric acid	HMDB0000714	others
167	Glycylproline	HMDB0000721	others
168	Hydroxyphenyllactic acid	HMDB0000755	others
169	Hydrocinnamic acid	HMDB0000764	others
170	Phenyllactic acid	HMDB0000779	others
171	Salicyluric acid	HMDB0000840	others
172	Phthalic acid	HMDB0002107	others
173	Imidazolepropionic acid	HMDB0002271	others
174	3-Indolepropionic acid	HMDB0002302	others
175	3-(3-Hydroxyphenyl)-3-hydroxypropanoic acid	HMDB0002643	others
176	N-Methylnicotinamide	HMDB0003152	others
177	gamma-Glutamylalanine	HMDB0006248	others
178	2-Phenylpropionate	HMDB0011743	others
179	Indolepyruvate	HMDB0060484	others
180	Butyric acid	HMDB0000039	SCFAs
181	Acetic acid	HMDB0000042	SCFAs
182	Propionic acid	HMDB0000237	SCFAs
183	Caproic acid	HMDB0000535	SCFAs
184	Isocaproic acid	HMDB0000689	SCFAs
185	Isovaleric acid	HMDB0000718	SCFAs
186	3-Hydroxyisovaleric acid	HMDB0000754	SCFAs
187	Valeric acid	HMDB0000892	SCFAs
188	Isobutyric acid	HMDB0001873	SCFAs
189	Ethylmethylacetic acid	HMDB0002176	SCFAs

BCAA: branched-chain amino acid;

SCFA: short-chain fatty acid

14) The Discussion should include a clear highlight in the first paragraph of the key take aways from the paper, from a high level, to guide the reader, given how much information is provided and the complexity of the findings. This should include both a) replication of prior findings and b) importantly, what is new and novel about the present work. Both of these categories of information should also be clear throughout the discussion section, for each finding cluster, as well as in the Conclusions.

Answer: Thank you for the suggestion. We have summarized key findings and revised the first paragraph of the discussion. Additionally, we have enhanced clarity throughout the discussion and conclusion sections to distinguish new findings from replications of prior results.

The revised first paragraph of the discussion is as follows:

In this study, we not only confirmed previously reported associations of conjugated BAs, BCAAs, and glutamate metabolism with AD progression but also extended our analysis to assess their performances in stratified populations and independent cohorts. Beyond their connections to clinical stages and cognitive function, we identified the association of glutamate-related features with AD pathology. Furthermore, we proposed the intermediary role of ammonia in the relationship between metabolic features and AD, providing additional support to the emerging hypothesis that ammonia disturbance contributes to AD progression. Our findings significantly contribute to expanding our comprehension of AD as a metabolic disease, offering observational evidence for early detection and advancing our understanding of AD pathology.

Revised conclusion:

In summary, leveraging the new Han cohort alongside several publicly available datasets, we validated the associations of conjugated BAs, BCAAs, and glutamate metabolism with AD. Our study unveiled the diverse performances of these metabolites in sub-populations, offering powerful evidence for the mechanistic link between metabolite-mediated ammonia abnormality and AD development. Future investigations are imperative to validate the clinical potentials of the identified features and unravel their interconnected roles in maintaining ammonia homeostasis, influencing A β deposition, and contributing to AD development. Further exploration in these directions holds promise for advancing our understanding and potential interventions in AD.

15) In the Discussion, the authors note that “In contrast, glutamate- and BCAA-related features were lower in AD and associated with better cognition.” Given the complexity of the findings, additional text should be provided to let the reader know what this exactly means - that is, for example, is having lower glutamate-related features in AD associated with better cognition in AD individuals only? What does this mean?

Answer: Thank you for the suggestions. Indeed, the interpretation is as you surmised: 'having lower glutamate-related features in AD is associated with better cognition in AD individuals.' We have undertaken substantial revisions in the descriptions of Figure 3, aligning with your comments and those of other reviewers (refer to the response to your question 8). In the revised version, this sentence has been omitted for clarity.

16) Improved discussion of the impact of differences in age, education, and so on across the subgroups on the outcomes needs to be added.

Answer: Thank you. In response to your suggestion, we have incorporated additional discussions (~500 words), along with 10 references. This expanded content delves into the potential impacts of age, sex, education, and *APOE-ε4* status on the association between metabolic profiles and the progression of AD. You can find these detailed discussions in the supplementary information section (SI-discussion).

Additional discussions added to the SI:

Potential Impacts of Age, Sex, Education Year, and *APOE-ε4* Status on the Association of Metabolic Profiles and AD Progression

The association of metabolic profiles with AD progression is intricately influenced by various factors. Factors such as age, sex, education year, and *APOE-ε4* status are recognized as significant contributors, often exerting greater impacts than other variables^{26, 27, 28}.

Age: Age, a primary non-modifiable risk factor for AD, is intricately linked to alterations in metabolic profiles during cognitive aging and AD development. These changes encompass dysregulated levels of BCAAs, modified bile acids, abnormal glutamine-glutamate cycle, impaired beta-amyloid clearance, and alterations in phosphatidylcholines (PCs) and sphingomyelins (SMs) in fatty acid composition and levels²⁹. Specific metabolites exhibit differential associations with cognitive decline and AD across various age groups. For instance, serum total cholesterol (TC) and low-density lipoprotein cholesterol (LDL-C) represent age-dependent risk factors for cognitive impairment among elderly participants, suggesting potential interactions between metabolites and age in AD²⁷. Metabolome-wide association studies support this, revealing several metabolite-by-age interactions significantly correlated with executive function, an early aspect of cognition affected during AD progression²⁸. The differential rates of metabolism for these particular metabolites could explain age-specific associations, necessitating compensatory mechanisms during younger years to account for rapid metabolism²⁸.

***APOE-ε4* Status:** The influence of *APOE-ε4* as a risk factor for AD has long been recognized, impacting cholesterol transport, brain lipid composition, beta-amyloid elimination, and neuroinflammation³⁰. Recently, it has emerged as a modifier of AD metabolism, with stratified analyses demonstrating *APOE-ε4*-dependent heterogeneous effects of metabolites on AD. Certain metabolites, such as PCs and proline, exhibit specific effects in female *ε4* carriers²⁶.

Sex: Sex-based modulation of the associations between metabolites and AD biomarkers has been substantiated, with direct evidence pointing to sex-dependent alterations in various metabolic pathways, including GABA synthesis, arginine biosynthesis, alanine, aspartate, and glutamate metabolism, fatty acid elongation, and lysophospholipid metabolism³¹. Stratified

analysis on ADNI data has highlighted substantial heterogeneity between sexes, emphasizing potential sex-specific interactions of metabolites and dysregulations in energy metabolism, energy homeostasis, and stress response. Notably, specific metabolic effects were identified in female $\epsilon 4$ carriers^{26, 27}.

Education: Education is linked to enhanced cognitive reserve and improved metabolic performance associated with AD, yet its potential influence on the association between metabolites and AD remains ambiguous³⁰. A Mendelian randomization study confirmed that the protective effect of education against AD is largely attributable to better cognition³². Individuals with higher education tend to participate in brain-stimulating activities and adopt healthier lifestyles, which are conducive to metabolic health³³.

Studies addressing the interplay of multiple factors on the association between metabolic profiles and AD are in their early stages. Distinct alterations in fatty acid metabolomics have been observed in *APOE- $\epsilon 4$* non-carriers and women, suggesting a nuanced role for *APOE- $\epsilon 4$* -sex intertwined effects in metabolic pathways relevant to AD³⁴. Age-related decreases in glutamate, GABA, and sphingolipids worsened with the increase of *APOE- $\epsilon 4$* load, potentially contributing to deficits in synaptic, learning, and memory-related functions³⁵. Interactions between sex and age have been underscored, supported by sex- and age-tailored correlations between serum lipids and cognitive impairment²⁷.

Reviewer #3 (Remarks to the Author):

In this paper, Chen and colleagues perform a metabolomic analysis in a cohort of individuals from the Chinese Preclinical Alzheimer's Disease Study (C-PAS). They first extracted 189 metabolic features from twelve different categories and used dimension reduction (PLS-DA, PCA) to identify multivariate differences between different diagnostic groups. They gradually work top-down to identify 34 specific features from the data and perform a series of linear and logistic regression analyses to determine whether there is a relationship between these features and diagnostic stage, neuropsychological assessment, and amyloid PET imaging (obtained in a subset of individuals). These features predominantly represent neurotransmitter activity (glutamate, glutamine, GABA), bile acids, and branch chain amino acids (BCAA). Substudies were performed stratifying by sex, *APOE- $\epsilon 4$* status, and age. Replication was done using the ADNI and ROSMAP studies. As these features have been found to be associated with elevated blood ammonia levels, an additional standard serum test around ammonia was performed in the C-PAS cohort, which increased with disease stage and showed association with BCAA and glutamate features.

The strength of this work is that multiple cohorts containing different ethnic makeups, provides more robust evidence that these markers are different at various stage of the disease. The

authors speak about using these markers as an early diagnostic tool. While identifying what biological basis there is for subjective cognitive decline, what I would be even more interested in whether these markers are elevated even earlier, in individuals who are cognitively normal but show signs of AD pathology.

Answer: Thank you for acknowledging the strength of our research. In response to your insightful query, we conducted additional analysis to explore the early performances of metabolic features by comparing their levels in individuals with normal cognition or with subjective cognitive decline (CN+SCD), stratified by A β positivity (n=40) and A β negativity (n=224). Our results revealed that GDCA/DCA was significantly (Mann-Whitney p=0.0082) higher in A β + individuals (n=40) compared to A β - individuals (n=224), though statistical significance was attenuated after adjusting for covariates (logistic regression adjusting age, sex, BMI, education year, and *APOE*- ϵ 4 status; p=0.073). This suggests early changes in gut microbiota deconjugation capability and bile acid pool structure in individuals with preclinical AD. Additionally, three glutamate-related features consistently showed lower levels, albeit not reaching statistical significance, in A β + compared to A β -. Further stratification analysis was not pursued due to the limited sample size, potentially missing features significant only in specific sub-populations. This new finding has been incorporated into the results section (2.4), addressing your major comment 1.

MAJOR COMMENTS

1. Overall: I'm having trouble understanding how the authors are placing these results in the context of what we as a field already know about the pathophysiology of the disease. Should these markers be considered a more fundamental element of AD pathology or a downstream marker of it? Are they specific to AD compared to other forms of dementia, or are they less specific to Alzheimer's disease itself and more helpful in terms of explaining heterogeneity within the disease, in terms of progression? This should be discussed a bit more in the manuscript.

Answer: Thank you for raising this crucial point. To better contextualize our results within the existing understanding of AD pathophysiology, we expanded our analysis to explore the associations between metabolic features and brain A β deposition—an established indicator of AD pathology. By comparing the levels of the 13 identified features between participants with positive and negative brain A β deposition, we aimed to elucidate whether these markers are fundamental elements of AD pathology or downstream markers of it. Our findings indicate that glutamate-related features and GDCA/DCA exhibit different levels between individuals with positive and negative brain A β deposition. This suggests a potential association of these markers with both the clinical syndrome and the underlying pathology of AD. Furthermore, voxel-wise analysis, stratified not only by age, sex, and *APOE*- ϵ 4 status but also by disease stage (CN + SCD vs. MCI + AD), was performed to provide refined evidence regarding the association of metabolic features with AD pathophysiology. However, it is important to note that

our current data do not allow us to determine the causal association of these features with A β deposition. Additionally, the specificity of these markers for AD remains uncertain, as our study did not include other forms of dementia for comparison. We acknowledge these limitations and have revised Section 2.4 and the discussions to reflect the nuances of our findings and the need for further exploration in understanding the specificity and causality of these metabolic features in the context of AD.

The revised results 2.4:

.....We observed consistent patterns in the levels of glutamate-related features across different populations. Specifically, glutamate-related features were consistently lower in individuals with positive A β (A β +) compared to those with negative A β (A β -) in CN+SCD groups (n=264), MCI+AD groups (n=157), and the entire study population (Figures 4a-4c). However, it's important to note that these differences did not reach statistical significance in the CN+SCD group alone, with p-values exceeding 0.05 in both the Mann-Whitney test and logistic regression adjusting for age, sex, BMI, education year, and *APOE*- ϵ 4 status. Despite this, we identified a noteworthy finding in CN+SCD subjects, where GDCA/DCA was significantly higher (Mann-Whitney p=0.0082) in A β + individuals (n=40) compared to A β - individuals (n=224). This suggests an early alteration in individuals without clinical symptoms but exhibiting signs of AD pathology. Unfortunately, the significance of GDCA/DCA was attenuated after adjusting for covariates (logistic regression adjusting for the aforementioned covariates; p=0.073).

Moreover, employing voxel-wise analysis, we delved into the associations between the identified features and brain A β deposition. Our results confirmed a negative association between glutamate-related features, particularly glutamate/glutamine, and brain A β deposition, predominantly in the frontal, lateral parietal, and lateral temporal lobes (Figures 4d-4t). This underscores a close correlation between the glutamate/GABA-glutamine cycle and AD pathology. Interestingly, this correlation exhibited some dependency on factors such as age, sex, *APOE*- ϵ 4 status, and disease stage.

The revised discussion (page 11, 2nd paragraph):

It is important to note that the current evidence is insufficient to fully determine whether this association is causal or a consequence of A β deposition. Additionally, as our study did not include other types of dementia, we cannot ascertain the specificity of these markers for AD. Further investigations are warranted to elucidate the interplay of glutamate-related features in both the brain and blood, unravel their causal associations, and determine their specificity in relation to brain A β deposition.

Revised Figure 4:

Figure 4. The associations of glutamate metabolism with brain Aβ deposition (n=421). The levels of glutamate (a), glutamate/glutamine (b), and glutamate/GABA (c) in CN+SCD, MCI+AD, and all participants stratified by positive and negative Aβ deposition which was determined by the consensus of physicians' visual interpretation of PET image. The solid line in violine plot represents the median and the dashed line represents quartile. * represents logistic regression p<0.05 adjusting age, sex, BMI, education year, and APOE-ε4. d)-t) the associations of glutamate-related features with Aβ, based on the voxel-wise analysis, in entire and age-/sex-/APOE-ε4/stage-stratified participants. The color bar represents the T value with the statistical threshold of p<0.05, peak-level FDR correction, and adjusted for age, sex, BMI, education year, and APOE-ε4 when applicable. d) Typical brain regions. Plasma levels of glutamate (e-g),

glutamate/glutamine (h-o), and glutamate/GABA (p-t) were negatively associated with brain A β deposition.

2. Line 125-128: Could the authors please indicate the rationale for generating the 34 features from the 29 metabolites. What criteria (statistical testing, prior literature?) was used in terms to include these features and exclude others?

Answer: Thank you. Based on the 29 metabolites (3 BCAAs, 8 excitatory neurotransmitters, and 18 bile acids), 34 extended features were generated and a total of 63 features were involved in subsequent analysis. All the 34 extended features are of clear biological meanings based on prior biological knowledge and literatures. We have added some references and listed the computational formula and meaning of the extended feature in Table S2.

The added references:

1. Jia W, Xie G, Jia W, et al. Bile acid-microbiota crosstalk in gastrointestinal inflammation and carcinogenesis. *Nat Rev Gastroenterol Hepatol*. 2018 Feb;15(2):111-128.
2. Jia W, Wei M, Rajani C, et al. Targeting the alternative bile acid synthetic pathway for metabolic diseases. *Protein Cell*. 2021 May;12(5):411-425.
3. Zheng P, Zeng BH, Xie P, et al. The gut microbiome from patients with schizophrenia modulates the glutamate-glutamine-GABA cycle and schizophrenia-relevant behaviors in mice, 2019 Feb 6;5(2):eaau8317. 10.1126/sciadv.aau8317
4. Cui M, Jiang YF, Zhao QH, et al. Metabolomics and incident dementia in older Chinese adults: The Shanghai Aging Study, *Alzheimer's Dement*. 2020 May;16(5):779-788. 10.1002/alz.12074

The revised Table S2:

Table S2. The computational formula and biological meaning of 34 extended metabolic features.

index	name	Biological meaning (KEGG K number or computational formula)	type
1	BCAAs	concentration of total BCAAs (valine+leucine+isoleucine)	summat ion
2	glutamate/GABA	activity of glutamate decarboxylase (K01580)	ratio
3	glutamate/glutamine	activity of glutamine synthetase, glutamate synthase (NADH), and glutaminase (K01915, K00264, and K01425)	ratio
4	asparate/N-acetyl-L-aspartate	activity of aspartate N-acetyltransferase and aspartoacylase (K18309 and K01437)	ratio

5	aspartate/asparagine	activity of asparagine synthase (glutamine-hydrolysing), aspartate--ammonia ligase, and glutamin-(asparagin-)ase (K01953, K01914, K05597)	ratio
6	glutamate/oxoglutarate	activity of glutamate synthase (NADPH) large chain (K00265)	ratio
7	TBA	total BAs (concentration summation of 19 BAs), plasma BA pool	summation
8	ConBA	total conjugated BAs (GCA+TCA+GCDCA+TCDCA+GDCA+TDCA+GUDCA+TUDCA+GLCA), BA pool structure	summation
9	UnconBA	total unconjugated BAs (CA+CDCA+DCA+UDCA+LCA), BA pool structure	summation
10	PriBA	total primary BAs (CA+CDCA+GCA+GCDCA+TCA+TCDCA), BA pool structure	summation
11	SecBA	total secondary BAs (DCA+UDCA+LCA+GDCA+TDCA+GUDCA+TUDCA+GLCA), BA pool structure	summation
12	Pri/Sec	ratio of primary and secondary BAs, BA pool structure	ratio
13	Con/Uncon	ratio of conjugated and unconjugated BAs, BA pool structure	ratio
14	CA+CDCA	total primary unconjugated BAs, BA pool structure	summation
15	CA/CDCA	ratio of 2 primary unconjugated BAs, balance of classical and alternative pathway of BA metabolism	ratio
16	TCA/CDCA	liver enzymatic (including the bile acid -CoA: amino acid N-acyltransferase, K00659 and sterol 12-alpha-hydroxylase, K07431) activities and gut microbiome function (bile salt hydrolase, K01442)	ratio
17	GCA/CDCA	liver enzymatic (including the bile acid -CoA: amino acid N-acyltransferase, K00659 and sterol 12-alpha-hydroxylase, K07431) activities and gut microbiome function (bile salt hydrolase, K01442)	ratio
18	DCA/CA	gut microbiome function (bile acid 7-alpha hydroxylation including K15868, K15870, K15872, K15871 and K15873)	ratio
19	GDCA/CA	liver enzymatic (bile acid -CoA: amino acid N-acyltransferase, K00659) activities and gut	ratio

		microbiome function (bile salt hydrolase, K01442; bile acid 7-alpha hydroxylation including K15868, K15870, K15872, K15871 and K15873)	
20	TDCA/CA	liver enzymatic (bile acid -CoA: amino acid N-acyltransferase, K00659) activities and gut microbiome function (bile salt hydrolase, K01442; bile acid 7-alpha hydroxylation including K15868, K15870, K15872, K15871 and K15873)	ratio
21	LCA/CDCA	gut microbiome function (bile acid 7-alpha hydroxylation including K15868, K15870, K15872, K15871 and K15873)	ratio
22	GLCA/CDCA	liver enzymatic (bile acid -CoA: amino acid N-acyltransferase, K00659) activities and gut microbiome function (bile salt hydrolase, K01442; bile acid 7-alpha hydroxylation including K15868, K15870, K15872, K15871 and K15873)	ratio
23	UDCA/CDCA	gut microbiome function (7alpha/beta-HSDH, K00076/K23231)	ratio
24	GUDCA/CDCA	liver enzymatic (bile acid -CoA: amino acid N-acyltransferase, K00659) activities and gut microbiome function (bile salt hydrolase, K01442; 7alpha/beta-HSDH, K00076/K23231)	ratio
25	TUDCA/CDCA	liver enzymatic (bile acid -CoA: amino acid N-acyltransferase, K00659) activities and gut microbiome function (bile salt hydrolase, K01442; 7alpha/beta-HSDH, K00076/K23231)	ratio
26	GLCA/LCA	gut microbiome function (bile salt hydrolase, K01442)	ratio
27	TCA/CA	gut microbiome function (bile salt hydrolase, K01442)	ratio
28	TCDCA/CDCA	gut microbiome function (bile salt hydrolase, K01442)	ratio
29	TDCA/DCA	gut microbiome function (bile salt hydrolase, K01442)	ratio
30	TUDCA/UDCA	gut microbiome function (bile salt hydrolase, K01442)	ratio
31	GCA/CA	gut microbiome function (bile salt hydrolase, K01442)	ratio
32	GCDCA/CDCA	gut microbiome function (bile salt hydrolase, K01442)	ratio
33	GDCA/DCA	gut microbiome function (bile salt hydrolase, K01442)	ratio
34	GUDCA/UDCA	gut microbiome function (bile salt hydrolase, K01442)	ratio

3. Section 2.4/Figure 4: For the cluster level FDR correction – the authors need to state what

the cluster forming threshold was used for this work, as a liberal cluster-forming threshold can be more sensitive to false positives.

Answer: Thank you for the suggestion. In our study, we used the peak-level FDR correction for voxel analysis, with a significance level set at 0.05 and the cluster-defining voxel threshold at the default of 0.001. This clarification has been added to the last paragraph of Section 5.6.

4. Section 2.4/Figure 4: The concern I have from Figure 4 is that this association is driven mostly by disease stage/severity, as most of the people with elevated amyloid PET are likely to be MCI or AD. It would be worth investigating how disease stage could be incorporated into the model, so regardless of disease severity (which has already been investigated in the other parts of the analysis), we can see what features are associated with elevated amyloid PET uptake.

Answer: We appreciate your observation. We recognize the importance of considering disease stage in our voxel-wise whole-brain association analysis. Subsequently, we conducted separate analyses for CN+SCD and MCI+AD participants, providing the results in Figures 4f, 4g, 4i, 4j, 4q, and 4r. These additional analyses confirmed the consistent association of glutamate-related features with brain A β deposition, highlighting that this association is partially dependent on age, sex, *APOE*- ϵ 4 status, and disease stage. Please refer to the response to your question 1 for the new figures and revised text.

5. Section 2.5: Was there any cross-validation performed in the ROC analysis to avoid overfitting?

Answer: Thank you for highlighting this crucial aspect. In response to your suggestion, we have incorporated a 7-fold cross-validation (CV) in the ROC analysis to mitigate overfitting. Figures 5a-5g have been revised accordingly. The text in Results (2.5), Methods (2nd paragraph in 5.6) and the legend of figure 5 have been updated to reflect this change. Notably, while the auROC improvements in CV were slightly lower compared to the self-test scenario, all improvements remained higher than zero. This consistency underscores the positive impact of metabolic features on AD diagnosis and prediction.

The revised Figure 5:

Figure 5. Metabolic features improved the performances of clinical markers in entire and stratified participants. The auROC values (mean with S.E. from 7-fold cross validation) of gradient boosting models using basic (age, sex, BMI, *APOE-ε4*, and education year; blue) and combined (basic and 13 metabolic features; red) features for the differentiation of every 2 stages, in all (a) and stratified (b-g) participants. h) Scatter plot of whole brain Aβ deposition level ([18F]florbetapir SUVR) and output of gradient boosting regression model with (red, Spearman correlation coefficient $r=0.47$, $p<0.0001$) and without (blue, $r=0.21$, $p<0.0001$) metabolic features.

The revised 2.5:

..... The inclusion of metabolic features consistently improved the auROCs (all higher than zero with an average of 0.05 and a maximum of 0.13) of models for stages differentiation and prediction (Figures 5a-5g).

6. Section 2.6: Multiple ADNI studies are listed as part of the meta-analysis, with a claimed total of 7,685 participants. However, there are only 2000 total participants across the various phases of ADNI. If you were to combine the sample sizes from the ADNI studies alone used in this manuscript, you would have a total of 7,119 samples, so there must be a fair amount of overlap between studies, and the question becomes if it's really surprising that similar coefficients would arise from data sets that share substantial amounts of data. The authors need to determine how many participants are overlapping between these studies and explain how they took this into account in their analysis of this data.

Answer: Thank you. Yes, considerable sample overlaps existed between the ADNI studies, as samples were often distributed to multiple centers for various measurements. These overlapping samples resulted in non-independent data, which could potentially lead to spurious positive findings. In response to this concern, we have undertaken several steps in the revision:

1. Provided detailed measurement information for all eight studies (Table S4), including instrument platform, team, qualitative or quantitative methods, and ADNI phase.
2. Presented the population characteristics of each study (Table S5).
3. Quantified the overlaps of samples between ADNI studies (Figure S4).

To address the issue of non-independence in the data, we re-analyzed the data using a three-level meta-analysis model (PMID: 23834422, ISBN 9780367610074; employing the `rma.mv` function in the `metafor` package). This meta-analysis method is specifically designed to account for dependencies between samples or data. As expected, the performances of the majority (10 out of 13) features were consistent with those observed in the C-PAS cohort. Figures 6, S3, and S4, along with related text and legends, have been updated accordingly. Detailed descriptions and references on the three-level meta-analysis method have been added to the SI-method section.

Revised results 2.6:

..... The performances of the 13 features were further verified in eight additional data sets comprising a total of 7,685 participants (Tables S4 and S5). Given the overlap of samples in ADNI data sets (Figure S4), we employed a three-level meta-analysis to evaluate their differences between CN and AD and their associations with global cognition. Detailed results for each feature are provided in Figures S5 and S6..... The majority (11 out of the 13) features showed significant alterations in AD compared to CN (Figure 6a, $p < 0.05$, random effect model), and their alteration trends were consistent with the findings from the C-PAS cohort. The overall results on Spearman correlation coefficients between metabolic features and global cognition scores (Figure 6b) indicated that 11 among the 13 features were associated with cognition ($p < 0.05$).

Revised Figure 6:

Figure 6. Three-level meta-analysis forest plots of identified features on their alteration trends with AD progression (a; mean difference with 95%CI of AD and CN stages; >0, higher in AD; <0, higher in CN) and on their associations with global cognition (b; Partial Spearman Rho value with 95%CI based on all available data. Age and sex were adjusted.). Cognition scores for ADNI, C-PAS, and ROSMAP data sets were ADAS-13, -1*MMSE, and a composite measure of global cognition created by averaging the z-scores of all tests respectively.

Revised Method 5.6 Statistical analyses:

For meta-analysis, given the overlap of samples from ADNI data set (Figure S4), the three-level meta-analysis model (random effect model), a method specifically designed to address dependencies between samples or data⁴³, was conducted to examine the standardized mean differences of features between CN and AD groups and their associations with global cognition (partial spearman correlation adjusting age and sex). Detailed information on dataset inclusion criteria, population characteristics, overlap of samples, and the three-level meta-analysis model are provided in supplementary information.

SI method for meta-analysis:

..... Statistical independence constitutes a fundamental assumption in meta-analysis when pooling effect sizes²². The presence of dependency between samples or data can artificially attenuate heterogeneity, potentially leading to spurious positive findings²³. This issue is known as the unit-of-analysis error. Considering the overlaps of samples in ADNI studies, we employed the three-level meta-analysis models on the identified features to evaluate their standardized mean differences between CN and AD ((mean level of ADs - mean level of CNs) / S.D. of CNs and ADs) and their correlations with global cognition (partial spearman correlation adjusting age and sex).

The three-level meta-analysis model is a meta-analysis method specifically designed to address dependencies between samples or data^{24, 25}. A three-level model consists of three levels of pooling. Initially, researchers combine the results of individual participants within their primary studies to report an aggregated effect size / correlation. Subsequently, at level 2, these effect sizes / correlations are nested within multiple clusters. Finally, pooling the aggregated cluster effects / correlations yields the overall true effect size μ .

Level 1 model:

$$\hat{\theta}_{ij} = \theta_{ij} + \epsilon_{ij}$$

Level 2 model:

$$\theta_{ij} = \kappa_j + \zeta(2)_{ij}$$

Level 3 model:

$$\kappa_j = \mu + \zeta(3)_j$$

Here, $\hat{\theta}_{ij}$ represents an estimate of the true effect size θ_{ij} , with ij indicating "effect size i nested in cluster j." The parameter κ_j denotes the average effect size within cluster j, while μ represents the overall average population effect. These formulas can be combined into a single line as follows:

$$\hat{\theta}_{ij} = \mu + \zeta(2)_{ij} + \zeta(3)_j + \epsilon_{ij}$$

This formula now encompasses two sources of heterogeneity: $\zeta(2)_{ij}$, signifying within-cluster heterogeneity at level 2, and $\zeta(3)_j$, representing between-cluster heterogeneity at level 3.

Consequently, fitting a three-level meta-analysis model necessitates estimating not only one heterogeneity variance parameter (τ^2) but also two (one for level 2 and another for level 3). We fitted the three-level meta-analysis models using the `rma.mv` function in the `metafor` package, employing maximum likelihood procedures²⁶.

7. Section 2.6: It is also not clear what the inclusion criteria for this replication study is, as well as what are the differences between the different studies, particularly the ADNI ones. This would be helpful information to go in the Supplementary Material.

Answer: Thank you. We have provided the inclusion criteria (SI method), measurement information (Table S4), population characteristics (Table S5) of the studies and the overlaps of samples in ADNI studies (Figure S4), per your suggestion.

SI method for inclusion criteria in meta-analysis section:

A total of 9 data sets were involved in the meta-analysis, including 7 from ADNI, 1 from ROSMAP, and the one from our C-PAS. These data sets had to meet the following criteria: 1) derived from peripheral blood samples; 2) from AD related studies with clinical or pathological diagnosis (containing at least CN and AD stages); 3) with at least one indicator of global cognitive function; 4) with age and sex information; 5) with test results for at least one of the 13 metabolic features; 6) related data are accessible.

Table S4: Metabolite measurement information of 8 validation data sets.

Data set name	Testing platform or assay	Testing team	qualitative / quantitative	ADNI phase
ADNI-Duke2016	The AbsoluteIDQ p180 assay	Duke University	quantitative	ADNI1
ADNI-Duke2017	The AbsoluteIDQ p180 assay	Duke University	quantitative	ADNI2, GO
ADNI-California2017	Gas chromatography time of flight mass spectrometry (GCTOFMS) instrument	University of California	qualitative	ADNI1
ADNI-Hawaii2021	Ultra-performance liquid chromatography coupled to tandem mass spectrometry (UPLC-MS/MS).	University of Hawaii Cancer Center	quantitative	ADNI1, GO, 2
ADNI-Nightingale2021	Nuclear Magnetic Resonance (NMR)	Nightingale Health's NMR metabolomics platform	quantitative	ADNI1, GO, 2

ADNI-DukeBAs2016	The Biocrates Bile Acids assay	Duke University	quantitative	ADNI1
ADNI-M2OVEAD2016	NA	NA	qualitative	NA
Rosmap-Hawaii2017	Ultra-performance liquid chromatography coupled to tandem mass spectrometry (UPLC-MS/MS)	University of Hawaii Cancer Center	quantitative	NA
C-PAS-Shanghai 2023	Ultra-performance liquid chromatography coupled to tandem mass spectrometry (UPLC-MS/MS)	University of Hawaii Cancer Center	quantitative	ADNI1, GO, 2

Table S5: Population characteristics of 8 validation data sets.

ADNI-Duke2016 (serum)	All (n=818)	CN (n=232)	LMCI (n=397)	AD (n=189)	
Age(yr)	75.23+6.83	75.89+5.06	74.78+7.45#	75.37+7.29	
Sex(M/F)	469/349*	119/113	257/140#	93/96#	
Education(yr)	15.53+3.02*	16.06+2.84	15.62+3.02	14.67+3.09#	
APOE(ε4+)%	37.53%*	24.14%	40.81%#	47.09%#	
ADAS13	18.48+9.29*	9.4+4.21	18.76+6.24#	29.02+7.56#	
ADNI-Duke2017 (serum)	All (n=898)	CN (n=182)	SMC (n=104)	EMCI/LMCI (n=474)	AD (n=138)
Age(yr)	72.46+7.28*	73.45+6.32	72.18+5.55	71.54+7.49#	74.55+8.3
Sex(M/F)	472/426*	88/94	45/59#	256/218#	83/55#
Education(yr)	16.27+2.62*	16.6+2.53	16.82+2.55	16.18+2.6#	15.75+2.77#
APOE(ε4+)%	35.75%*	24.73%	31.73%	37.76%#	46.38%#
ADAS13	15.42+9.65*	9.07+4.5	8.69+4.09	14.76+6.8#	31.12+8.54#
ADNI-California2017 (serum)	All (n=820)	CN (n=232)	LMCI (n=398)	AD (n=190)	
Age(yr)	75.23+6.82	75.89+5.06	74.79+7.44#	75.35+7.28	
Sex(M/F)	470/350*	119/113	258/140#	93/97#	
Education(yr)	15.53+3.02*	16.06+2.84	15.62+3.01	14.68+3.09#	
APOE(ε4+)%	37.44%*	24.14%	40.70%#	46.84%#	
ADAS13	18.46+9.28*	9.4+4.21	18.74+6.25#	28.97+7.58#	
ADNI-Hawaii2021	All (n=1172)	AD(n=186)	CN(n=350)	EMCI/LMCI	

(serum)	(n=636)				
Age(yr)	73.90+7.11*	75.30+7.70	74.82+5.71	72.98+7.49#	
Sex(M/F)	649/523*	101/85#	181/169	367/269#	
Education(yr)	15.94+2.84*	14.98+2.99#	16.36+2.74	15.99+2.79#	
APOE(ε4+)%	36.95%*	25.43%#	39.62%	49.46%#	
ADAS13	16.26+8.99*	28.97+7.76#	9.11+4.21	16.48+6.78#	
ADNI-Nightingale Health2021 (serum)	All (n=1681)	CN (n=404)	SMC (n=104)	EMCI/LMCI (n=854)	n=AD (n=319)
Age(yr)	73.78+7.21*	74.78+5.81	72.18+5.55#	73.05+7.63#	75.30+7.70
Sex(M/F)	917/764*	202/202	45/59#	500/354#	170/149#
Education(yr)	15.93+2.83*	16.31+2.72	16.82+2.55	15.93+2.78#	14.98+2.99#
APOE(ε4+)%	36.59%*	24.26%	31.73%	39.23%#	46.71%#
ADAS13	16.82+9.62*	9.16+4.30	8.69+4.09	16.53+6.86#	28.98+7.76#
ADNI- DukeBAs2016 (serum)	All (n=833)	AD (n=191)	CN (n=233)	EMCI/LMCI (n=399)	
Age(yr)	74.35+10.64*	75.36+7.26	75.94+5.11	74.80+7.44#	
Sex(M/F)	473/360*	94/97#	120/113	259/140#	
Education(yr)	15.35+3.45*	28.97+7.56#	9.43+4.23	15.63+3.02	
APOE(ε4+)%	36.97%*	47.12%#	24.03%	40.60%#	
ADAS13	18.25+9.43*	28.97+7.56#	9.43+4.23	18.73+6.25#	
ADNI- M2OVEAD2016 (serum)	All (n=897)	AD (n=138)	CN (n=182)	EMCI/LMCI (n=473)	SMC (n=104)
Age(yr)	72.47+7.28*	74.55+8.30	73.45+6.32	71.55+7.49#	72.18+5.55
Sex(M/F)	472/425*	83/55#	88/94	256/217#	45/59#
Education(yr)	16.27+2.62*	15.75+2.77#	16.60+2.53	16.18+2.61	16.82+2.55
APOE(ε4+)%	35.67%*	46.38%#	24.73%	37.63%#	31.73%
ADAS13	15.42+9.66*	31.12+8.54#	9.07+4.50	14.767+6.806#	8.69+4.09
Rosmap- Hawaii2017 (serum)	All (n=566)	AD (n=13)	CN (n=446)	MCI (n=107)	
Age(yr)	82.28+7.49*	86.12+8.88#	81.13+7.38	86.65+6.02#	
Sex(M/F)	447/119	10/3#	356/90	81/26	
BMI	27.61+5.48*	24.80+6.71	28.01+5.50	26.23+4.95#	
Education(yr)	15.68+3.05*	17.15+3.76	15.71+3.06	15.26+2.85	
Global cognition	0.19+0.59	-1.23+0.77	0.38+0.44	-0.41+0.48	
APOE(ε4+)%	17.49%*	30.77%	16.14%	22.12%	

Data are presented as mean+S.D., percentage, or number. * indicates Chi-squared test, analysis of variance, or Kruskal–Wallis test FDR<0.05 comparing 4 groups (adjusted by Benjamini and Hochberg). # indicates Chi-squared test, student’s t-test or Mann-Whitney test FDR<0.05 compared to CN (adjusted by Benjamini and Hochberg). CN: cognitively normal; AD: Alzheimer’s disease; SCD: subjective cognitive decline; MCI: mild cognitive impairment; EMCI: early MCI; LMCI: late MCI; SMC: significant memory concern.

Figure S4:

Figure S4. The number of overlapped participants between ADNI studies.

8. Lines 327-331: This is not sufficient enough information around the PET image processing. It states that “A consensus of physicians visually interpreted the PET images.” But then there is voxelwise analysis and the use of a continuous measure in the Figures. I’m unclear where the visual reads came into play. While there is some more information in the Supplementary, it either needs to go in the main manuscript or it needs to be referenced in the main text. There are also more details required in terms of the reference region and the target region when performing a global measure such as in Figure 5(h).

Answer: Thank you for your valuable feedback. We have enhanced the information regarding PET image processing in the main manuscript (Section 5.3) and Supplementary Information (SI-Method). Specifically, we clarified that the determination of positive and negative Aβ deposition status was based on the physicians' visual interpretation of PET images, and this information was used in Figures 4a-4c. For the voxel-wise association analysis (Figures 4d-4t), we explored the associations between metabolic features and Aβ deposition. In Figure 5h, the global Aβ deposition level (x-axis) was measured as the whole-brain Aβ standardized uptake value ratio (SUVR) using cerebellum grey matter as the reference region. Global cortical amyloid

burden was calculated as the average SUVR in 7 cortical area, including posterior cingulate, precuneus, frontal, lateral parietal, lateral temporal, medial temporal, and occipital regions.

The revised method 5.3:

5.3 Quantitative calculation and visual interpretation of PET images

[18F]Florbetapir PET data were quantified using the standardized uptake value ratio (SUVR), with the cerebellum grey matter serving as the reference region. The global cortical A β burden was computed as the mean SUVR in cortical area, including posterior cingulate, precuneus, frontal, lateral parietal, lateral temporal, medial temporal, and occipital regions.

Positive [18F]-florbetapir PET images were defined through visual rating, following the guidelines for interpreting amyloid PET³⁹. Three physicians independently assessed all amyloid PET images, and results were determined based on a consensus, with agreement among at least two physicians. Additional details on the acquisition and preprocessing of [18F]Florbetapir PET neuroimaging are provided in the supplementary information.

The revised method in SI:

2. Brain PET neuroimaging acquisition and preprocessing

[18F]Florbetapir PET/CT scans were employed to evaluate A β plaques in the brain. The tracer was produced in the Department of Nuclear Medicine & PET Center, Huashan Hospital, Fudan University, adhering to Good Manufacturing Practice (GMP) conditions.

PET/CT imaging was performed using PET/CT scanners (Biograph mCT Flow, Siemens, Erlangen, Germany) with parameters described previously^{16, 17}. Twenty-minute scans were conducted 50 minutes post-injection of approximately ~37 MBq/kg ($\pm 10\%$) of [18F]florbetapir intravenously. Following acquisition, the PET images underwent reconstruction using a filtered back-projection algorithm with corrections for decay, normalization, dead time, photon attenuation, scatter, and random coincidences.

PET image preprocessing was conducted using SPM12 (Wellcome Trust Centre for Neuroimaging, London, UK; <https://www.fil.ion.ucl.ac.uk/spm>) and CAT12 (<http://www.neuro.uni-jena.de/cat>) following a previously outlined procedure^{18, 19}. After reorienting PET and T1-weighted MR images, PET images were co-registered to individual T1-weighted images. Subsequently, the T1-weighted images were segmented using CAT12, and the generated tissue-labeled images were utilized for partial volume correction (PVC) of PET images employing the Muller-Gartner method²⁰. Then the deformation field file from segmentation was used to transform corresponding PET images into the MNI space, and finally images were smoothed using a Gaussian filter with a full width at half of the maximum (FWHM) equal to 8 mm. The global cortical A β burden was computed using the preprocessed images in MNI space, represented as the mean SUVR in cortical area, including posterior cingulate, precuneus, frontal, lateral parietal, lateral temporal, medial temporal, and occipital regions.

MINOR COMMENTS

1. Line 99-101: "The majority of participants were women (65.19%), and 18.51% were identified as *APOE-E4* positive (Table 1)." When I look at Table 1, it indicates that the positivity rate was 28.88%. Please could you explain the discrepancy?

Answer: Thank you. The 18.51% in the main text refers specifically to the percentage of female *APOE-ε4* carriers, whereas the 28.88% in Table 1 represents the percentage of *APOE-ε4* carriers, encompassing both men and women, in the full population. To eliminate confusion, we have adjusted the main text to state: 'The majority of participants were women (65.19%), and 28.88% were identified as *APOE-ε4* positive (Table 1).'

2. Table 1: Please reduce the number of significant figures to make the data easier to read.

Answer: Thank you. We have retained one significant digit in Table 1.

3. Table 1: In addition to how many participants from each group had a PET scan, the number of individuals that were classified as amyloid positive through visual read is important information to include.

Answer: Thank you. We have added this information in Table 1.

4. Please replace the term gender with sex in the manuscript.

Answer: Thank you. We have replaced all the gender with sex.

5. There is no section 4 in the manuscript.

Answer: Thank you. We have revised the section numbers and section 4 is the conclusion.

6. Figure 2/Results: For each of the metabolite PCA's, it would be helpful to provide what percentage of variation in the data that each PC1 is capturing. This could be put into the Figure itself or somewhere in the text.

Answer: Thank you. The percentage of variation each PC1 captured has been provided in Figures 2c and 2d.

Revised Figures 2c and 2d:

Figure 2. BAs, BCAAs, and excitatory neurotransmitters exhibit strong associations with AD. c) Fold changes of PC1s (the first component of PCA) derived from the metabolite types in subjects with SCD, MCI, and AD relative to CN. * indicates ANOVA FDR<0.05 when comparing NC, SCD, MCI, and AD. \$ indicates post hoc Dunnett's test p<0.05 when compared to CN. Metabolite types are arranged in decreasing order of ANOVA FDR values. The number next to the name represents the percentage of variation that PC1 captured. d) Levels of five sub-types belonging to the top three types of figure 2c in four clinical stages. The levels are represented by the PC1 scores derived from PCA based on metabolites belonging to each sub-type and are fitted by LOESS. The number next to the name represents the percentage of variation that PC1 captured.

7. Figure 5(h): the x axis label is vague. It is not clear what measurement of PET is actually being performed here; it should be clear whether this is SUVR and ideally what reference region. Answer: Thank you for the comment. This is SUVR, and the cerebellum grey matter was used as the reference region. We have added this information to the figure legend, and the x-axis has been revised to 'global [¹⁸F]florbetapir SUVR.

Reviewer #4 (Remarks to the Author):

Mass spectrometry-based metabolomics is a powerful tool for analyzing human plasma in biomarker discovery. In this manuscript, the authors employed the targeted metabolic method to analyze a total of 189 metabolites in a large-scale study encompassing over 1300 Chinese AD cohort cases. This cohort is well-documented with cognitive and neuropsychological tests, complemented by PET scans using the ¹⁸F-AV-45 tracer in 421 cases. The authors managed to correlate certain metabolite profiles with clinical data, reinforcing their findings through validation with 8 published datasets from ADNI and ROSMAP. Notably, they emphasized the discovery of elevated blood ammonia levels in AD cases. This study leverages a unique dataset

and highlights potential future biomarker targets. Specific concerns are as follows:

1. The study reports a unique, extensive Chinese cohort and a wealth of clinical and metabolomic data, despite comparable studies previously carried out by ADNI and ROSMAP.

Answer: Thank you for the positive comments on our cohort study.

2. The finding of increased ammonia levels in AD patients has been previously reported and reviewed. It slightly diminishes the novelty of this manuscript.

Answer: Thank you. Indeed, the increased ammonia levels in AD patients have been previously reported, as acknowledged in the manuscript. Our contribution lies in the confirmation of elevated ammonia levels in a large Chinese Han cohort and in establishing connections between the changes in metabolic features and the increased ammonia levels.

3. It's imperative that all pertinent raw files, such as MS raw data and individual metabolite measurements, be made readily accessible. The current supplementary data lacks this information, even though it refers to an external website for data deposition (<http://www.metabolomicsatlas.com>). Details regarding how to access this data should be clarified.

Answer: Thank you. The MS raw data and individual metabolite measurements have been made readily accessible at MetaboLights (accession No. MTBLS4554).

The revised data availability: "..... Metabolomics data of C-PAS cohort are accessible at MetaboLights (accession No. MTBLS4554)."

4. The manuscript cites an Anal Chem paper for the targeted approach using UPLC-MS/MS to measure 189 metabolites. The discussion should address the pros and cons of this method and elucidate the criteria for selecting these specific metabolites to substantiate the dataset's reliability.

Answer: Thank you for the suggestion. The pros and cons of the Anal Chem method, developed by our group in 2021, are now addressed in the SI-method, as per your recommendation. This high-throughput system enables the simultaneous determination of approximately 200 metabolites across various classes.

The revised method:

"..... This automatic and high-throughput system allows for the simultaneous determination of as many metabolites as possible, approximately 200, spanning various classes. We constructed a combined MS library of 3-NPH derivatives from structurally diverse compounds to facilitate metabolite identification. The system demonstrated excellent linearity, reproducibility, and stability, making it suitable for large clinical applications. However, it's important to note that, like any technology, there are limitations. The coverage of certain metabolite classes, such as many lipid classes, is limited to ensure stability and accuracy....."

In total, 199 metabolites were measured, and 10 that fell below the limit of quantification were excluded from subsequent analysis.”

5. The section on data processing seems incomplete, requiring additional details on outlier removal, batch correction, and the examination of potential confounding factors.

Answer: Thank you for the suggestion. We have added more information of data processing in Methods section (5.4 and 5.6, see below).

The raw data files underwent processing using TMBQ software (V1.0, HMI, Shenzhen, China), encompassing peak integration, calibration, quantification, quality control, and batch effect adjustment for each metabolite, adhering to the manufacturer’s guidelines. Outliers were identified through Cauchy distribution robust fit ($K \sigma=7$). Any outliers (<0.2%) and missing values (<0.1%) were substituted using multivariate normal imputation. To normalize their distribution for statistical analysis, the data underwent logarithmic transformation (base=2).

The determination of potential covariates for adjustment involved balancing the need to prevent confounding while limiting model complexity.

6. To enhance the practical application of the findings, it is advisable that the authors pinpoint a small panel of metabolites to propose as promising biomarker targets.

Answer: Thank you for the suggestion. We have proposed a panel of 4 features as promising biomarker targets as their overall performances were better than the others and were linked to different biological significance. This has been addressed in discussion.

“We here propose a small panel of 4 features as promising biomarker targets considering their overall performances and underlying biological significance. They are GDCA/DCA, GDCA/CA, valine, and glutamate/glutamine.”

Reviewers' Comments:

Reviewer #1:

Remarks to the Author:

The authors have been very responsive to the comments from the four Reviewers. Significant and appropriate changes have been made to the original manuscript. The issues as identified have all been addressed and this has emerged as a strong paper reporting important findings.

Reviewer #2:

Remarks to the Author:

The authors were very responsive to reviewer suggestions and concerns, and the manuscript and study have been improved and clarified. The addition of the age-matched analyses are very helpful, and the provision of an overarching summary in the beginning of the discussion is as well. The addition of greater detail for the complex figures is important for the reader, despite the length. The only minor comment is regarding Table 1. Table 1 has the additional range and IQR information, which is very helpful, however, the table could be made more readable by including the range and IQR within the same cell as the mean+sd, they don't need separate rows and as presented make it difficult to find the variables. The definition of the range[] and IQR can be mentioned in the Table legend/footnote.

Reviewer #3:

Remarks to the Author:

The authors have put considerable thought and effort into addressing the reviewer comments. Most of the points that I have raised have been addressed. I still have some minor queries and clarifications that would be helpful.

1. The authors have added "All participants in the C-PAS study had to fulfill the following criteria: (1) Age between 40 and 85 years with a minimum education duration of more than 1 year...", yet in Table 1 there is a minimum age in the control group of 31? Could you explain how they could be included in the study?
2. Thank you for adding the number of participants with A β PET, as well as the number of those individuals who were A β +. It is interesting that the NC, SCD individuals have A β rates roughly in line with other studies, but I find that only 20% of MCI and 52% of the AD individuals who had a PET scan being A β + to be quite low. Could the authors please comment on this? Is there a risk that some of these individuals classified as AD have different forms of dementia or that the impairment is caused by some other cause other than AD pathology?
3. The extremely young age at onset for some of the AD patients suggests that they either carry an autosomal dominant mutation or they were misdiagnosed. Did the authors do any genetic screening for the youngest AD individuals in their group to determine if any of them carried one of these mutations?
4. Re-examining Figure 2(d), I'm not sure I understand what fitting has been done? Is this a LOESS fit for each metabolite class based on the four data points (NC, SCD, MCI, AD) or are there more points involved (or does it involve all five classes of metabolite in one model. Looking at the plot again, these LOESS fits don't appear to fit very well to some of the underlying points.
5. Centroid is misspelled in Figure 2b
6. "Positive associations (with positive effect sizes) were observed for BA-related features" – As another reviewer noted, there is a lot of information to convey to the reader in this section and Figure 3, so I would just be very precise about what these associations are with. Are you saying BA-related features are positively associated with clinical stage? Disease severity?
7. The subgroup analysis is particularly interesting in terms of the substantial difference in associations according to sex and APOE status. It makes me wonder if this highlights some potential different pathways depending on if an individual is APOE+ or APOE-.
8. "between every two stages (Figure 3g)", I think that would be clearer if it said "in pairwise comparison between clinical stages"
9. On Figure 4 (e)-(t) – I think the layout should be reconsidered. The layout is fine for the first two rows which look at glutamate in the first row and glutamate/glutamine in the second row, but

then in subsequent rows that further breakdown by subgroup in different metrics, some metabolite features come in mid-row, and it difficult to follow.

10. The Glutamate/GABA plot at the bottom of Figure S5 looks a bit different than the rest.

11. Could the authors please clarify in the text what pooling was done as part of the three-level meta-analysis? I'm assuming this pooled the ADNI data together (or at the very least the datasets with high overlap), while ROSMAP and C-PAS are in another pool?

12. "Unfortunately, the significance of GDCA/DCA was attenuated after adjusting for covariates..." I don't think that it is fortunate or unfortunate when assessing whether a p-value goes above or below a threshold

13. Figure S4 – perhaps instead of total number of overlapping patients, you could show the percentage of overlapping patients with respect to the overall number of unique subjects between both studies (i.e Intersection over Union). This would make it easier to compare overlap between studies.

Reviewer #4:

Remarks to the Author:

All of my previous concerns have been addressed. The manuscript is acceptable for publication.

Reviewer #1 (Remarks to the Author):

The authors have been very responsive to the comments from the four Reviewers. Significant and appropriate changes have been made to the original manuscript. The issues as identified have all been addressed and this has emerged as a strong paper reporting important findings.

Answer: Thank you for the positive response.

Reviewer #2 (Remarks to the Author):

The authors were very responsive to reviewer suggestions and concerns, and the manuscript and study have been improved and clarified. The addition of the age-matched analyses are very helpful, and the provision of an overarching summary in the beginning of the discussion is as well. The addition of greater detail for the complex figures is important for the reader, despite the length. The only minor comment is regarding Table 1. Table 1 has the additional range and IQR information, which is very helpful, however, the table could be made more readable by including the range and IQR within the same cell as the mean+sd, they don't need separate rows and as presented make it difficult to find the variables. The definition of the range [] and IQR can be mentioned in the Table legend/footnote.

Answer: Thank you for the positive response. We have revised the format of Table 1 according to your suggestion (please refer to the answer to reviewer 3 question 1).

Reviewer #3 (Remarks to the Author):

The authors have put considerable thought and effort into addressing the reviewer comments. Most of the points that I have raised have been addressed. I still have some minor queries and clarifications that would be helpful.

1. The authors have added "All participants in the C-PAS study had to fulfill the following criteria: (1) Age between 40 and 85 years with a minimum education duration of more than 1 year...", yet in Table 1 there is a minimum age in the control group of 31? Could you explain how they could be included in the study?

Answer: Thank you for bringing this to our attention. Upon thorough review of the enrollment criteria, we identified an oversight in the inclusion of two women in the CN group who did not meet the specified age criteria, being aged 31 and 36. Consequently, these individuals have been excluded from the study. We have adjusted the inclusion age range from "Age between 40 and 85 years" to "Age between 40 and 90 years." Subsequently, all relevant sections, including text, tables, and figures, have been updated to accurately reflect these amendments.

Table 1. Characteristics of study population from C-PAS cohort

Characteristics	ALL (n=1397)	CN(n=487)	SCD(n=239)	MCI(n=284)	AD(n=387)
-----------------	--------------	-----------	------------	------------	-----------

Age (yr)	66.2+8.6* [40, 89] 66 (60, 72)	63.6+8.2 [40, 84] 64 (58, 69)	64.4+7.4 [47, 81] 64 (58, 70)	66.5+8.1# [43, 86] 66 (61, 73)	70.4+8.4# [41, 89] 71 (65, 77)
Sex (Men%)	34.4%	33.7%	28.9%	34.5%	38.8%
BMI (kg/m ²)	23.3+3.8 [13.7, 33.8] 23.2 (21.0, 25.4)	23.4+3.4 [15.4, 33.8] 23.3 (21.2, 25.3)	24.0+3.5 [16.4, 31.6] 23.7 (21.5, 26.0)	23.1+3.3 [15.5, 33.2] 23.0 (20.9, 25.4)	22.9+3.4 [13.7, 31.1] 22.7 (20.6, 25.2)
Education (yr)	11.4+3.2* [6, 22] 11 (9, 14)	12.4+3.1 [6, 22] 12 (10, 15)	11.9+3.1 [6, 20] 12 (9, 14)	11.1+3.0 [6, 22] 11 (9, 12)	10.2+3.2# [6, 19] 10 (7, 12)
APOE (ε4) carrier % ^a	28.8%*	19.1%	17.6%	29.6%#	47.6%#
PET acceptance(%) ^b	30.1%*	34.8%	39.3%	31.7%#	17.3%#
Brain Aβ+(%) ^c	28.5%	15.3%	18.1%	31.1%	73.1%#
MMSE	24.6+5.6* [10, 30] 27 (22, 29)	28.2+1.7 [20, 30] 28.5 (27, 29)	27.7+1.8# [21, 30] 28 (26, 29)	26.5+2.1# [15, 30] 27 (25, 28)	16.8+4.7# [10, 27] 17.5 (12, 21)
ACEIII-CV	68.8+18.2* [10, 97] 73 (60, 82)	82.0+7.9 [60, 97] 83 (77, 88)	77.8+8.0# [60, 96] 78 (73, 83)	70.3+9.0# [50, 94] 71 (64, 76)	45.7+14.7# [10, 77] 48 (36, 58)
MoCA-BC	23.3+4.8* [10, 30] 24.50 (20, 27)	26.1+2.5 [20, 30] 27 (25, 28)	24.7+3.0# [17, 30] 25 (22, 27)	21.9+3.4# [15, 30] 22 (20, 24)	15.2+3.3# [10, 22] 15 (12, 18)

Data are presented as mean+S.D., [minimum, maximum], and median (IQR), or percentage. * indicates Chi-squared test, analysis of variance, or Kruskal–Wallis test FDR<0.05 when comparing 4 groups (adjusted by Benjamini and Hochberg). # indicates Chi-squared test, student's t-test or Mann-Whitney test FDR<0.05 when compared to CN (adjusted by Benjamini and Hochberg). C-PAS: Chinese Preclinical Alzheimer's Disease Study; CN: cognitively normal; AD: Alzheimer's disease; SCD: subjective cognitive decline; MMSE: Mini-Mental State Examination; ACEIII-CV: Chinese version of Addenbrooke's cognitive examination-III; MoCA-BC: Chinese version of Montreal Cognitive Assessment-Basic. a: the percentage of APOE-ε4 carriers. b: the percentage of the participants that accepted brain PET test. c: the percentage of participants with positive Aβ (defined through visual rating following the guidelines for interpreting amyloid PET) in those underwent the brain AV45-PET scans.

2. Thank you for adding the number of participants with Aβ PET, as well as the number of

those individuals who were A β +. It is interesting that the NC, SCD individuals have A β + rates roughly in line with other studies, but I find that only 20% of MCI and 52% of the AD individuals who had a PET scan being A β + to be quite low. Could the authors please comment on this? Is there a risk that some of these individuals classified as AD have different forms of dementia or that the impairment is caused by some other cause other than AD pathology?

Answer: We appreciate your thoughtful observation, and upon further examination, we acknowledge that there was an error in the grouping of patients based on A β positivity in the previous version of the manuscript. The discrepancy arose from the use of a cutoff value of [18F]florbetapir standardized uptake value ratio (SVUr) to define A β positivity. In this revised version, we have rectified this by adopting the consensus approach as we indicated in method 5.3, wherein the classification of participants as A β positive or negative is based on visual assessments by three independent physicians interpreting amyloid PET readings. We believe this methodology aligns more closely with current standards within the AD clinical community.

As a result of this adjustment, the revised A β positivity rates for clinically diagnosed individuals are as follows: 15.3% for NC, 18.1% for SCD, 31.1% for MCI, and 73.1% for AD. These rates are consistent with findings from our previous multi-centered study on amyloid positivity in the urban Chinese population, of which a substantial portion of participants were from our C-PAS cohort. In that study, the A β positivity rates for CN, MCI, and AD were 26.9%, 34.6%, and 78.3%, respectively [1]. We have updated the data in Table 1 (please refer to the answer to your first question for details). Importantly, this correction only affects the A β positivity rates in Table 1, and no other changes have been made to the related results and text in the article.

[1] He K, Li B, Huang L, Zhao J, Hua F, Wang T, et al. Positive rate and quantification of amyloid pathology with [(18)F]florbetapir in the urban Chinese population. *European radiology*. 2023.

3. The extremely young age at onset for some of the AD patients suggests that they either carry an autosomal dominant mutation or they were misdiagnosed. Did the authors do any genetic screening for the youngest AD individuals in their group to determine if any of them carried one of these mutations?

Answer: Thank you for your insightful comments, and we appreciate your concerns regarding the unusually young age at onset observed in some AD patients in our study. Recognizing the higher possibility of pathogenic genes associated with familial AD in these cases, we conducted genetic testing for mutations in APP, PS1, and PS2 in a subset of AD patients who demonstrated good compliance. Approximately 3% of these individuals were found to carry mutations, with around half of them being under the age of 60. While it is recognized that autosomal dominant AD may result in elevated brain A β burden and plasma levels of A β 42 and A β 40 compared to sporadic AD, the associations with metabolites and ammonia remain less explored.

Due to the limited number of individuals with autosomal dominant AD genes, a comprehensive stratified analysis on this subgroup is challenging. Nevertheless, we have

validated our results in a sub-population aged older than 60 years, as suggested by reviewer 2. This information has been transparently included in the limitations section of our study, where we acknowledge the potential inclusion of more patients with familial AD and the relative youthfulness of our subjects. We emphasize that despite these considerations, the identified features consistently perform well in other cohorts with older subjects and fasting samples, bolstering the reliability of our findings.

Additionally, we want to assure you of the accuracy of our clinical and PET diagnoses. The clinical department and PET center responsible for these assessments are esteemed teams in the field, certified by the Chinese Ministry of Health. Their staff undergo extensive, specialized training, adhering strictly to relevant standards and criteria. We believe this commitment to excellence contributes to the accuracy of our diagnoses.

4. Re-examining Figure 2(d), I'm not sure I understand what fitting has been done? Is this a LOESS fit for each metabolite class based on the four data points (NC, SCD, MCI, AD) or are there more points involved (or does it involve all five classes of metabolite in one model. Looking at the plot again, these LOESS fits don't appear to fit very well to some of the underlying points.

Answer: Indeed, the fitting in Figure 2(d) involves a LOESS fit for each metabolite class based on the four data points (NC, SCD, MCI, AD). Upon reconsideration, we acknowledge that the LOESS fits may not align perfectly with some of the data points, such as the 2nd and 3rd points of BCAAs, due to significant variation. In our efforts to find the most suitable fitting method, we explored various linear and nonlinear approaches, including centered 2nd, 3rd, and 4th-order polynomials, 1st, 2nd, and 3rd-phase decay, cumulative Gaussian, sigmoid, exponential, and logistic growth models. Ultimately, more sophisticated methods yielded improved fitting results.

The choice of LOESS was influenced by several factors: 1) the limited number of points to be fitted, 2) the preference for a simple trajectory, and 3) the widespread use and familiarity of LOESS in similar analyses. We have incorporated additional details on the generation of trajectories for metabolite types in Method Section 5.6, specifically in the second paragraph on page 15. Furthermore, we have referenced two recent publications wherein the same strategy was employed to depict metabolic and microbial trajectories, providing additional context to our approach.

The added method reads as follows:

"The levels of metabolite types were represented by the first principal components (PC1s) derived from principal component analysis (PCA) based on metabolites of corresponding types. Locally weighted regression (LOESS) was used for curve fitting of PC1s ^{43,44}."

43. Zhou L, Chen TL, Jia W et al. Effects of vaginal microbiota transfer on the neurodevelopment and microbiome of cesarean-born infants: A blinded randomized controlled trial. *Cell Host & Microbe*, 31, 1232-1247 (2023), 10.1016/j.chom.2023.05.022.

44. Chen T, Jia W et al. Serum bile acids improve prediction of Alzheimer's progression in a sex-dependent manner. *Advanced Science*, Online ahead of print, (2023), 10.1002/advs.202306576.

5. Centroid is misspelled in Figure 2b

Answer: Thank you. This word has been corrected.

6. "Positive associations (with positive effect sizes) were observed for BA-related features" – As another reviewer noted, there is a lot of information to convey to the reader in this section and Figure 3, so I would just be very precise about what these associations are with. Are you saying BA-related features are positively associated with clinical stage? Disease severity?

Answer: Thank you for your feedback. We have revised the sentence to provide greater precision. The revised sentence now reads as follows: "Positive associations (positive effect sizes) with disease severity were observed for BA-related features, whereas BCAA and glutamate-related features displayed negative associations (negative effect sizes) with disease severity." (bottom of page 5).

7. The subgroup analysis is particularly interesting in terms of the substantial difference in associations according to sex and APOE status. It makes me wonder if this highlights some potential different pathways depending on if an individual is APOE+ or APOE-.

Answer: Thank you for your insightful comment. We wholeheartedly agree with your observation regarding the intriguing differences in associations based on sex and APOE status in the subgroup analysis. In our discussion, we have explicitly emphasized this point, stating: "This implies that employing distinct sets of metabolic features tailored to specific stages and populations enhances the precision in characterizing the metabolic patterns of AD. These observations may also help explain the inconsistencies in associations between metabolites and AD reported in other studies, and aid in interpreting the differences in pathophysiology and symptoms observed across different populations."

Furthermore, to provide a more comprehensive exploration of these potential pathways, we have expanded our discussion and included additional references in the supplementary information. This additional content delves into the potential influences of age, sex, and APOE- ϵ 4 status on the association between metabolic profiles and the progression of AD. It's worth noting that the impact of APOE- ϵ 4 on the metabolome of AD stands as one of our future research interests. We appreciate your keen observation, and your insights have enriched the depth of our discussion.

8. "between every two stages (Figure 3g)", I think that would be clearer if it said "in pairwise comparison between clinical stages"

Answer: Thank you. This sentence has been revised as you suggested (the second paragraph of page 6).

9. On Figure 4 (e)-(t) – I think the layout should be reconsidered. The layout is fine for the first two rows which look at glutamate in the first row and glutamate/glutamine in the

second row, but then in subsequent rows that further breakdown by subgroup in different metrics, some metabolite features come in mid-row, and it difficult to follow.

Answer: Thank you for your thoughtful suggestion. In response to your feedback, we have reorganized the layout of Figure 4 for improved clarity. Specifically, we have now grouped results for the same feature and the same metrics together. The variable "glutamate/glutamine," which generated 8 images, is now presented in the first three rows. The other two variables, "glutamate/GABA" and "glutamate," generated 5 and 3 images, respectively, are now arranged in the last two rows. We believe this rearrangement will enhance the visual coherence and ease of interpretation for the readers.

Figure 4. d) Typical brain regions. Brain A β deposition were negatively associated with glutamate/glutamine (e-l), glutamate/GABA (m-q), and glutamate (r-t), based on the voxel-wise analysis, in entire and stage-/sex-/age-/APOE- ϵ 4-stratified participants. The color bar represents the T value with the statistical threshold of $p < 0.05$, peak-level FDR correction, and adjusted for age, sex, BMI, education year, and APOE- ϵ 4 when applicable.

10. The Glutamate/GABA plot at the bottom of Figure S5 looks a bit different than the rest.

Answer: Thank you. We have updated the format of Glutamate/GABA plot according to

the others of Figure S5.

11. Could the authors please clarify in the text what pooling was done as part of the three-level meta-analysis? I'm assuming this pooled the ADNI data together (or at the very least the datasets with high overlap), while ROSMAP and C-PAS are in another pool?

Answer: We conducted a pooled analysis that combined data from all the studies involved, including ADNI, ROSMAP, and C-PAS. For the three-level meta-analysis, in addition to the conventional meta-analysis information, we specified which studies had duplicate samples by providing names with the same prefix. This step was crucial to address the non-independence of studies introduced by duplicate samples. It is important to note that the three-level meta-analysis is specifically designed to handle situations with duplicate samples, offering a robust theoretical foundation and utilizing a well-established R package.

While we explored an alternative approach with a two-step meta-analysis, wherein the first meta-analysis incorporated studies from ADNI and the second meta-analysis used the combined result along with data from ROSMAP and C-PAS, the outcomes were comparable to those of the three-level meta-analysis. Our preference for the three-level meta-analysis was influenced by its tailored design for situations involving duplicate samples.

To provide additional clarity on the pooling process, we have included a clarification in the Supplementary Information (SI): "All applicable data sets and the independence between them (which datasets have duplicate samples) were entered at once."

12. "Unfortunately, the significance of GDCA/DCA was attenuated after adjusting for covariates..." I don't think that it is fortunate or unfortunate when assessing whether a p-value goes above or below a threshold.

Answer: We agree with you and have removed the word "unfortunately".

13. Figure S4 – perhaps instead of total number of overlapping patients, you could show the percentage of overlapping patients with respect to the overall number of unique subjects between both studies (i.e Intersection over Union). This would make it easier to compare overlap between studies.

Answer: Thank you. We have revised the figure per your suggestion.

Figure S4 The percentage of overlapping patients with respect to the overall number of unique subjects between both studies (i.e Intersection over Union).

Reviewer #4 (Remarks to the Author):

All of my previous concerns have been addressed. The manuscript is acceptable for publication.

Answer: Thank you for the positive feedback.

Reviewers' Comments:

Reviewer #3:

Remarks to the Author:

The authors have adequately addressed all my queries, with one minor exception. Thanks to the authors for clarifying the LOESS curve fit in Figure 2(d). However, I don't think it adds anything for the reader, and it is likely not an appropriate method to fit just four data points, where each point represents a summary of a clinical group that still contains substantial variability. I think it would be better just to display the points without the fits myself.

Reviewer #3 (Remarks to the Author):

The authors have adequately addressed all my queries, with one minor exception. Thanks to the authors for clarifying the LOESS curve fit in Figure 2(d). However, I don't think it adds anything for the reader, and it is likely not an appropriate method to fit just four data points, where each point represents a summary of a clinical group that still contains substantial variability. I think it would be better just to display the points without the fits myself.

Answer: Thank you for the thoughtful suggestion. We have revised Figure 2d and related text accordingly.

Figure 2. BAs, BCAAs, and excitatory neurotransmitters exhibit strong associations with AD. d) Levels (mean with S.E.) of five sub-types belonging to the top three types of Figure 2c in four clinical stages. The levels are represented by the PC1 scores derived from PCA based on metabolites belonging to each sub-type and were scaled to the same starting point. The number next to the name represents the percentage of variation that PC1 captured.